# EGEA-DM: EIGENVALUE-GUIDED EXPLAINABLE AND ACCELERATED DIFFUSION MODEL

## ABSTRACT

Diffusion models have achieved remarkable success in generating high-quality data, yet challenges remain in training convergence, interpretability, and fine-grained controllability. Additionally, the high computational cost of training is often overlooked from a theoretical perspective. To address these limitations, we propose Eigenvalue-Guided Explainable and Accelerated Diffusion Model (EGEA-DM), a novel framework grounded in ergodic theory. EGEA-DM leverages the $L$-generator's principal eigenvalue to control the forward diffusion speed, enabling adaptive adjustment of reverse steps during both training and sampling. By modulating the forward process through the $L$-generator's coefficients, our method establishes a unified mechanism for explainable and fine-grained control. This control, in turn, enables more efficient training and allows for the optimization of the speed-quality trade-off. Extensive experiments validate the effectiveness of EGEA-DM, demonstrating its potential to advance the practical applicability of diffusion models.

## 1 INTRODUCTION

Diffusion models are a powerful class of generative models that have achieved outstanding performance in various fields, such as image synthesis (Dhariwal & Nichol, 2021), audio generation (Huang et al., 2023), and time-series prediction (Shen & Kwok, 2023). These models operate by employing a forward process that iteratively adds noise to the data, gradually transforming the data distribution into a stationary distribution. The reverse process is then learned to progressively denoise the data, reconstructing the original distribution.

Diffusion models primarily fall into two categories: Denoising Diffusion Probabilistic Models (DDPM) (Ho et al., 2020) and Score-Based Generative Models (SGM) (Song et al., 2020c). DDPM-based methods, such as FastDPM (Kong & Ping, 2021), Truncated Diffusion Models (Zheng et al., 2022), and ES-DDPM (Lyu et al., 2022), optimize noise scheduling, truncation, and sampling efficiency. SGM employs score functions with stochastic differential equations (SDEs) or probability flow ODEs, with advancements like Lévy Stable Diffusion (Song & Zhang, 2023), MSGM (Liu & Wang, 2024), and adaptive step-size methods (Franzese et al., 2023), enhancing flexibility, robustness, and efficiency. These advancements collectively highlight the versatility and potential of diffusion models in addressing complex generative tasks while motivating further exploration into their theoretical foundations and practical applications.

Despite their major advances and impressive capabilities, diffusion models face two key challenges: *1) Theoretical gaps in diffusion sampling:* Diffusion models typically involve a large number of time steps (often exceeding 1000), which leads to significant computational overhead—especially during sampling—due to the need for repeated evaluations of a neural network in the reverse denoising process. While these models have shown impressive empirical performance, the theoretical understanding of this inefficiency and how to mitigate it remains limited. *2) Lack of interpretability and controllability:* While various methods (Kim et al., 2025; Fu et al., 2025; Jiang et al., 2024) etc. have been proposed to mitigate the computational cost of diffusion models, many lack a solid theoretical foundation. This limits their interpretability and constrains fine-grained control over the diffusion process, ultimately hindering systematic optimization and adaptation for diverse applications.

To address the computational inefficiency and lack of theoretical interpretability in diffusion models, we propose the Eigenvalue-Guided Explainable and Accelerated Diffusion Model (EGEA-DM), a novel framework grounded in ergodic theory. In EGEA-DM, we model the forward diffusion process using a continuous-time Markov generator governed by an $L$-generator. The convergence rate toward the stationary distribution is mainly determined by the spectral gap of this generator, which is equal to the magnitude of its principal (first non-zero) eigenvalue.

We modulate the coefficients of the $L$-generator to control the spectral decay, enabling fine-grained regulation of the forward diffusion dynamics. By estimating the deviation between the data distribution at step $T$ and the stationary distribution under a given generator, we determine the minimal number of forward (and hence reverse) steps required for effective denoising, thereby reducing the computational cost of both training and sampling.

Crucially, we observe that aggressive acceleration—i.e., maximizing the principal eigenvalue—can degrade generation quality due to insufficient representation of intermediate data states. To balance quality and efficiency, we incorporate empirical quality metrics into the eigenvalue-guided tuning process. This yields an interpretable trade-off curve between diffusion speed and generation fidelity.

A central technical challenge lies in estimating the principal eigenvalue of the $L$-generator. We adopt Chen's estimation theory (Chen, 2012), combined with iterative numerical methods, to efficiently approximate this quantity. This allows us to characterize and control the diffusion process via a theoretically grounded mechanism.

We remark that, although the eigenvalue estimation incurs additional computational overhead, it is substantially outweighed by the training acceleration achieved. Moreover, while other factors such as initial and stationary distributions do affect model convergence, the spectral properties exhibit dominant influence. Ultimately, while alternative theories can characterize convergence rates, they generally lack precise estimation bounds for generic diffusion processes.

Overall, EGEA-DM provides a principled and explainable approach to accelerating diffusion models, achieving joint optimization of computational efficiency and generative quality through spectral control. *Our contributions are summarized as follows:*

- **Interpretable diffusion via egodic theory:** We reinterpret diffusion models through the lens of ergodic theory, linking the convergence rate of the forward process to the principal eigenvalue of the $L$-generator. This provides a theoretical foundation for understanding and analyzing diffusion dynamics and noise injection schemes.
- **Controllable optimization via $L$-generator modulation:** By adjusting the coefficients of the $L$-generator based on its principal eigenvalue, we introduce a flexible mechanism to control the speed and stability of the diffusion process.
- **Spectral characterization via numerical estimation:** We adopt the iterative method (Chen, 2012) to efficiently estimate the principal eigenvalue of the $L$-generator, enabling quantitative control of diffusion speed and providing a metric for generator design.
- **Generalization across dataset and methods:** EGEA-DM demonstrates strong generalization on multiple datasets and integrates seamlessly with a variety of DDPM extensions, validating its robustness across tasks and architectures. Our framework is also compatible with score-based models, offering a theory-informed and systematic methodology approach for selecting and tuning $L$-generators across diverse generative frameworks, opening up new possibilities for expanding research on baseline models.

## 2 Preliminary

We briefly review score-based generative models (SGMs) and the associated $L$-generator. Here we focus on the one-dimensional case without loss of generality. In fact, although the experimental data distribution is high-dimensional and contains both semantic and spatial information, each dimension undergoes noise injection and removal independently according to the same Stochastic Differential Equations (SDEs), and thus shares the same $L$-generator, stationary distribution and principle eigenvalue. The only distinction lies in the potentially different initial distributions across dimensions. Consequently, the convergence rate of the high-dimensional data can be characterized mainly by this identical eigenvalue.

## 2.1 SCORE-BASED GENERATIVE MODEL

In SGMs, the forward diffusion process gradually perturbs data $X_0$ into noise, with the distribution of the final data $X_T$ at time $T$ approaching a stationary distribution. This process is governed by the following SDE:

$$\mathrm{d}X_t = f(X_t, t)\,\mathrm{d}t + g(X_t, t)\,\mathrm{d}W_t, \tag{1}$$

where $W_t$ is a standard Wiener process, $f(x, t) : \mathbb{R} \times [0, T] \to \mathbb{R}$ is the drift coefficient dictating the deterministic dynamics, and $g(x, t) : \mathbb{R} \times [0, T] \to \mathbb{R}$ is the diffusion coefficient scaling the random noise at each step.

The reverse process starts from samples of $X_T$ and iteratively denoises the data to recover $X_0$. This is described by the reverse-time SDE:

$$\mathrm{d}X_t = \left[ f(X_t, t) - g(X_t, t)^2 \nabla_x \log p_t(X_t) \right] \mathrm{d}t + g(X_t, t)\,\mathrm{d}\widetilde{W}_t,$$

where $\widetilde{W}_t$ is a standard Wiener process when time flows backward from $T$ to 0, and $\nabla_x \log p_t(x)$ is the score function, i.e., the gradient of the log-probability density of $X_t$ (typically unknown). $\nabla_x \log p_t(x)$ could be learned via sliced score matching (SSM), where a neural network is trained to approximate the gradients of perturbed data distributions across multiple noise scales. SSM trains the score matching function $s_\theta(X_t, t)$ by the following equaiton

$$\theta^* = \arg\min_\theta \mathbb{E}_t \left\{ \lambda(t) \mathbb{E}_{X_0} \mathbb{E}_{X_t} \mathbb{E}_{v \sim \mathcal{N}(0,1)} \left[ \frac{1}{2} \|s_\theta(X_t, t)\|_2^2 + \mathrm{Tr}\left( \nabla_{X_t} s_\theta(X_t, t) \right) \right] \right\},$$

where the random vector $v$ follows a Gaussian distribution, and $\lambda : [0, T] \to \mathbb{R}_{>0}$ is a weight function, taken as $\lambda \propto 1/\mathbb{E}\left[ \|\nabla_{X_t} \log p_{0t}(X_t \mid X_0)\|_2^2 \right]$ (Song et al., 2020c;b).

## 2.2 $L_t$-GENERATOR

In addition to the SDE in Eq. 1, the forward diffusion process can be fully characterized by the infinitesimal generator $L_t$-generator, defined as $L_t\,\phi(x) = \lim_{h\to 0} \frac{\mathbb{E}[\phi(X_{t+h})\mid X_t=x] - \phi(x)}{h}$, where $\phi$ is an infinitely differentiable function with compact support (Stroock & Varadhan, 1997). This generator specifies the evolution of $X_t$ at each infinitesimal time step. Using Itô's formula, the $L_t$-generator can be expressed as:

$$L_t = \frac{1}{2} g^2(t, x) \frac{\mathrm{d}^2}{\mathrm{d}x^2} + f(t, x) \frac{\mathrm{d}}{\mathrm{d}x}, \tag{2}$$

which encapsulates both the drift and diffusion components of the process. Note that if $\frac{1}{2} g^2(t, x) \equiv a(x)$ and $f(t, x) \equiv b(x)$ for all $t$, then $L_t$ is time-independent in $t$.

# 3 EGEA-DM

This section will present the detailed formulation of our proposed EGEA-DM. We adopt the SGM as the foundational framework and employ the $L$-generator to regulate the model. The adoption of the SGM is grounded in its rigorous theoretical guarantees for score matching, while the $L$-generator serving as an effective controller of the diffusion model.

Subsection 3.1 will develop the theoretical model design, including the generator structure, convergence conditions and principle eigenvalue-convergence rate correspondence. Subsection 3.2 will provide the numerical method to estimate the convergence speed, enabling model adjustment guided by the principle eigenvalue. Subsection 3.3 will present the empirical observations for selecting the $L$-generator, thereby enabling effective control of EGEA-DM.

## 3.1 ERGODIC THEORY

For EGEA-DM, the forward diffusion process $X_t$ is designed to satisfy the SDE

$$\mathrm{d}X_t = \beta(t) b(X_t)\mathrm{d}t + \sqrt{2\beta(t) a(X_t)}\mathrm{d}W_t, \tag{3}$$

so that its $L_t$-generator has the form

$$L_t = \beta(t) \cdot L \quad \text{with} \quad L = a(x)\frac{\mathrm{d}^2}{\mathrm{d}x^2} + b(x)\frac{\mathrm{d}}{\mathrm{d}x}, \tag{4}$$

where $a(x) > 0$ and $b(x)$ are both first-order continuous functions on $\mathbb{R}$, the scheduling function $\beta(t)$ is integrable and $0 < \beta_{\min} \le \beta(t) \le \beta_{\max} < \infty$.

Note that, once $\beta(t)$ is selected, the generation and properties of $X_t$ will be determined by $L$.

Before demonstrating the correlation between the convergence rate of $X_t$ and the principle eigenvalue of $L$, we should ensure the solution uniqueness and ergodicity of $X_t$. Solution uniqueness implies the convergence of $X_t$, while ergodicity guarantees that $X_t$ converges almost surely to a positive stationary distribution. The beolw Theorem 1 specifies what conditions on $a(x) > 0$ and $b(x)$ ensure both uniqueness and ergodicity of $X_t$ .

**Theorem 1** (Uniqueness and Ergodicity). *Given $X_0$, the solution $X_t$ of Eq. 3 is unique and ergodic if and only if*

$$\kappa(+\infty) = +\infty = \kappa(-\infty), \quad Z := \int_{\mathbb{R}} \frac{e^{C(u)}}{a(u)} du < +\infty, \tag{5}$$

*where $\kappa(x) = \int_0^x e^{-C(u)} \int_0^u \frac{e^{C(v)}}{a(v)} dv du$ and $C(x) = \int_0^x \frac{b(u)}{a(u)} du$ for $x \in \mathbb{R}$. If Eq. 5 holds, then the stationary distribution is $\pi(dx) = \frac{1}{Za(x)} e^{C(x)} dx$ for $x \in \mathbb{R}$.*

Let $L^2(\pi)$ be the real measure space $\{f : \pi(f^2) < \infty\}$ equipped with the norm $\|f\| = [\pi(f^2)]^{1/2}$ and the inner product $(f, g) = \int_{\mathbb{R}} f(x)g(x)\pi(dx)$, where $\pi(g) = \int_{\mathbb{R}} g(x)\pi(dx)$ for general $g$. The principle eigenvalue $\lambda_1$ of $L$ is defined as

$$\lambda_1 = \inf\left\{(f, -Lf) : f \in \mathscr{D}(L), \pi(f) = 0, \|f\| = 1\right\}, \tag{6}$$

with $\mathscr{D}(L)$ the domain of $L$ in $L^2(\pi)$. Since $L$ has one trivial eigenvalue $\lambda_0 = 0$, the spectral gap $\lambda_1 - \lambda_0$ is equal to $\lambda_1$ (Chen & Mao, 2021).

For any $B \in \mathcal{B}$, define $P_t(B) = \mathbb{P}\{X_t \in B\}$, where $\mathcal{B}$ is the collection of all Borel sets on $\mathbb{R}$. Then $P_t$ characterizes the distribution of $X_t$, while the stationary distribution $\pi(B) = \int_B \pi(dx)$. The total variation distance between $P_t$ and $\pi$ is defined as $\|P_t - \pi\|_{\text{Var}} = \sup_{B \in \mathcal{B}} |P_t(B) - \pi(B)|$.

The theorem below describes the convergence rate of $X_t$ toward to $\pi$.

**Theorem 2** (Convergence Rate). *Under the condition in Eq.5 of Theorem 1, it holds that*

$$\|P_t - \pi\|_{Var} \le \|P_0 - \pi\|_{Var} e^{-\lambda_1 \int_0^t \beta(s)ds}.$$

From Theorem 2, the convergence rate of $X_t$ increases monotonically with the magnitude of $\lambda_1$, through which we could precisely control the speed of the forward diffusion process. While $P_0$ and $\pi$ do affect convergence speed, $\lambda_1$ dominates the long-term dynamics, which is also shown in our experiments in Section 4.

However, $\lambda_1$ is typically difficult to obtain with exact precision, which explains why the next subsection performs its numerical estimation.

### 3.2 ESTIMATION OF THE PRINCIPAL EIGENVALUE

This subsection outlines an iterative algorithm for numerically estimating $\lambda_1$, based on Chen's theoretical estimation for this principle eigenvalue (Chen, 2012).

Initialize a function $f_1(z)$ as follows:

$$f_1^{x,y}(t) = \begin{cases} \left[ \dfrac{\int_y^r \frac{e^{c(u)}}{a(u)}\mathrm{d}u \cdot \int_{-\infty}^z \frac{e^{c(u)}}{a(u)}\mathrm{d}u}{\int_{-\infty}^x \frac{e^{c(u)}}{a(u)}\mathrm{d}u} \right]^{1/2}, & \text{if } z \leq x, \\[20pt] \left[ \displaystyle\int_y^\infty \frac{e^{c(u)}}{a(u)}\mathrm{d}u \right]^{1/2}, & \text{if } x \leq z \leq y, \\[20pt] \left[ \displaystyle\int_t^\infty \frac{e^{c(u)}}{a(u)}\mathrm{d}u \right]^{1/2}, & \text{if } z \geq y. \end{cases}$$

For the $n$-th step, define

$$f_n(z) = \begin{cases} f_n^-(z), & \text{if } z \leq \theta_n; \\ f_n^+(z) & \text{if } z > \theta_n, \end{cases}$$

where

$$f_n^-(z) = \int_{-\infty}^z \frac{e^{c(u)}}{a(u)}\mathrm{d}u \cdot \int_u^{\theta_n} e^{-c(t)} f_{n-1}^{x,y}(t)\mathrm{d}t, \quad f_n^+(z) = \int_z^{+\infty} \frac{e^{c(u)}}{a(u)}\mathrm{d}u \cdot \int_{\theta_n}^u e^{-c(t)} f_{n-1}^{(x,y)}(t)\mathrm{d}t,$$

and $\theta_n$ is obtained by solving the below equation for every $(x,y)$:

$$\int_{-\infty}^{\theta_n} \frac{e^{c(u)}}{a(u)}\mathrm{d}u \cdot \int_u^{\theta_n} e^{-c(t)} f_{n-1}^{x,y}(t)\mathrm{d}t = \int_{\theta_n}^{+\infty} \frac{e^{c(u)}}{a(u)}\mathrm{d}u \cdot \int_{\theta_n}^u e^{-c(t)} f_{n-1}^{(x,y)}(t)\mathrm{d}t.$$

Then the $n$-th estimation for the $\lambda_1$ is

$$\lambda_1^{(n)} = \inf_{x<y} \left[ \sup_z \frac{f_n^-(z)}{f_{n-1}(z)} \right] \vee \left[ \sup_z \frac{f_n^+(z)}{f_{n-1}(z)} \right]. \tag{7}$$

By Chen's theory, this sequence asymptotically and increasingly approaches to the true value of $\lambda_1$ as $n$ tends to $+\infty$.

In practical computations, we approximate the integral using the classical rectangle method, and the segmentation is 2000 intervals. For derivable cases, for instance, in DDPM, the error is approximately 0.0007. The computational time is about 2 hours. However, the calculation is performed on an Intel Core i5-9300H processor using MATLAB. Hence, if executed on hardware with higher specifications as model training and generation, the time is expected to be significantly reduced.

This method ensures tractable eigenvalue estimation, enabling the design of tuning the $L$-generator in the following subsection. We remark that, Eq. 7 demonstrates that convergence rate regulation requires eigenvalue estimation, as mere tuning of hyperparameters $a(x)$ and $b(x)$ yields suboptimal control accuracy.

### 3.3 How to Choose the $L$-Generator

It can be seen from Eq. 7 that the magnitude of the principal eigenvalue of the L-generator is determined by $a(x)$ and $b(x)$. Therefore, we can regulate the $L$-generator by selecting different forms of $a(x)$ and $b(x)$ under the guidance of this eigenvalue. This paper considers the case where both $a(x)$ and $b(x)$ are polynomial functions. It is a relatively common form both in theory and application. And, other common continuous functions can be approximated by polynomial functions, which is guaranteed by the Weierstrass approximation theorem (Stone, 1948). The selection of $L$-generator could be guided by the following three (empirical) principles:

**Principle I :** As theoretically analyzed in Subsection 3.1, $a(x)$ and $b(x)$ need to fulfill Eq. 5, ensuring that the forward diffusion process can converge to a positive stationary distribution. Eq. 5 can be readily verified through numerical experiments. Based on this verification, we have summarized the characteristics of relevant functions in 7 and 8 in Appendix .

**Principle II:** The following Theorem 3 reveals how the principle eigenvalue varies when linear transformations are applied to $a(x)$ and $b(x)$, providing another guiding principle for choosing the $L$-generator.

**Theorem 3.** *Let $a^*(x) = ka(x)$ and $b^*(x) = kb(x)$, where $k$ is a positive constant. Then the principle eigenvalue $\lambda_1^*$ of $L^* = a^*(x)\frac{d^2}{dx^2} + b^*(x)\frac{d}{dx}$ is equal to $k\lambda_1$.*

**Observation I:** According to Subsection 3.2, we have computed the eigenvalues corresponding to multiple $(a, b)$ pairs, as detailed in Tables 1-5. These computed values can serve as references and are amenable to minor adjustments, since the eigenvalues change continuously with $(a, b)$ under certain conditions—a fact corroborated by Kato-Rellich theorem (Kato, 2013). Figure 5 in the Appendix C.1 shows the evolution law of the principal eigenvalue as the degrees of $x$ in the polynomials $a(x)$ and $b(x)$ increase, providing a more intuitive visual representation.

## 4 EXPERIMENTS

We conduct experiments to evaluate the impact of the $L$-generator, especially its principal eigenvalue, on diffusion model performance.

### 4.1 EXPERIMENT SETUP AND EVALUATION METRICS

**Experiment setup.** Following the theoretical principles established in last section, we instantiate the $L$-generator by selecting appropriate functions $a(x)$ and $b(x)$. These choices give rise to different diffusion processes and spectral properties, corresponding to specific instances of our proposed EGEA-DM framework.

We implement our model using a U-Net architecture. All training processes were conducted on an NVIDIA GeForce RTX 4090 GPU. Key training parameters included: an initial learning rate of $1 \times 10^{-5}$. The model was optimized using the Adam optimizer with default momentum parameters ($\beta_1 = 0.9$, $\beta_2 = 0.999$). For fair comparison, we retain the linear, uniformly increasing noise schedule $\beta_t$ used in the original DDPM framework (Ho et al., 2020). We evaluate performance on two standard image generation benchmarks: CIFAR-10 (Krizhevsky, 2009), CelebA-HQ (Gábor Mélyi & Felippo, 2020), Image-NetDeng et al. (2009) 128×128 and 256×256.

**Evaluatation metrics.** To assess both the generation quality and acceleration efficiency of EGEA-DM, we employ four evaluation metrics: *1) Fréchet Inception Distance (FID) (Tim Salimans, 2016)*. A standard metric that measures the distance between real and generated image distributions. Lower FID indicates better visual quality and diversity. *2) Convergence Discrepancy ($D_{disc}$)*. $D_{\text{disc}}$ quantifies the proximity between the forward distribution $P_t(x)$ and the stationary distribution $\pi$: $D_{\text{disc}} = \|P_t - \pi\|_{\text{Var}} \approx \frac{1}{2}\sum_{\Delta x_i}|P_t(\Delta x_i) - \pi(\Delta x_i)|$ functioning as the guidance indicator for $T_{\text{conv}}$ determination.Lower $D_{\text{disc}}$ means smaller error between the reverse sampling process and the training distribution at time $T$. This is because the training phase draws samples from the distribution $P_T$, while the sampling phase operates based on the distribution $\pi$. *3) Step Count $T_{conv}$*. $T_{\text{conv}}$ is the required noise injection steps to make $D_{\text{disc}}$ suffciently small, indicating the convergence speed. *4) Training Time ($T_{spend}$)*. The total wall-clock time to train the model, used for comparing computational efficiency across methods.

### 4.2 EFFICIENCY GAINS VIA EIGENVALUE CONTROL

From Theorem 2, the eigenvalue $\lambda_1$ of the $L$-generator should be directly correlated with the model's training convergence rate. Specifically, a larger eigenvalue will require fewer training iterations (or less time) to reach the same loss threshold. The experimental results confirm this prediction.

Based on the analysis in Subsection 3.3, two configurations are considered: *1) Fixing $a(x)$ with different orders and varying the coefficient of $b(x)$ (Tables 1-4); 2) Fixing eigenvalues with varying $a(x)$ and $b(x)$ (Table 5).* The results reveal the following pattern.

**Observation II:** From Tables 1-5, on the same dataset, the greater the eigenvalue, the fewer steps ($T_{conv}$) and the less training time ($T_{spend}$) are required to achieve convergence discrepancy ($D_{\text{disc}}$). And similar eigenvalues incur comparable time costs. These both demonstrate that the eigenvalue predominantly govern the model's convergence rate, which is consistent with the Theorem 2.

Table 1: $L$-generators with fewer steps on CIFAR-10

| $a(x)$ | $b(x)$ | Eigenvalue | $T_{\text{conv}}$ | $D_{\text{disc}}$ | FID (SDE solver) | FID (NFE=15) (dpm solver) | FID (NFE=15) (dpm solver++) | $T_{spend}$ |
|---|---|---|---|---|---|---|---|---|
| $\frac{1}{2}$ | -0.25x | 0.24 | 1000 | 0.241 | 4.76 | 4.89 | 4.76 | 52h |
| $\frac{1}{2}$ | -0.5x | 0.48 | 1000 | 0.208 | 4.44 | 4.59 | 4.44 | 45h |
| $\frac{1}{2}$ | -x | 1.03 | 825 | 0.209 | **3.15** | 3.74 | **3.15** | 35h |
| $\frac{1}{2}$ | -2x | 2.04 | 750 | 0.208 | 3.19 | **3.23** | 3.19 | 30h |
| $\frac{1}{2}$ | -5x | 6.15 | 525 | 0.209 | 4.30 | 4.43 | 4.30 | 26h |
| $\frac{1}{2}$ | -10x | 10.70 | 350 | 0.208 | 6.33 | 6.55 | 6.33 | **20h** |

Table 2: $L$-generators with fewer steps on CelebA-HQ-64

| $a(x)$ | $b(x)$ | Eigenvalue | $T_{\text{conv}}$ | $D_{\text{disc}}$ | FID (SDE solver) | FID (NFE=20) (dpm solver) | FID (NFE=20) (dpm solver++) | $T_{spend}$ |
|---|---|---|---|---|---|---|---|---|
| $\frac{1}{2}$ | -0.25x | 0.24 | 1000 | 0.242 | 4.02 | 4.29 | 4.06 | 136h |
| $\frac{1}{2}$ | -0.5x | 0.48 | 1000 | 0.208 | 3.82 | 3.95 | 3.90 | 104h |
| $\frac{1}{2}$ | -x | 1.03 | 800 | 0.209 | **3.34** | **3.88** | **3.54** | 67h |
| $\frac{1}{2}$ | -2x | 2.04 | 750 | 0.209 | 3.67 | 3.96 | 3.73 | 61h |
| $\frac{1}{2}$ | -5x | 6.15 | 500 | 0.210 | 5.03 | 5.71 | 5.34 | 51h |
| $\frac{1}{2}$ | -10x | 10.70 | 300 | 0.209 | 7.41 | 8.09 | 7.99 | **45h** |

## 4.3 GENERATION PERFORMANCE UNDER EIGENVALUE GUIDANCE

Tables 1-5 show that generation quality is primarily governed by three factors: model complexity (as reflected in the functional forms of $a$ and $b$), the eigenvalue, and the dataset. The findings exhibit the following regularity.

**Observation III:** From Tables 1-5, greater model or dataset complexity achieves lower FID at a relatively slower convergence rate, while under similar complexity and fixed dataset, generation quality — dominated by the eigenvalue — follows a concave trend characterized by an initial FID decrease succeeded by an increase beyond an eigenvalue threshold.

The phenomena described above are readily explicable. Increased model complexity amplifies the data-dependent variability of SDE (Eq. 3) coefficients, raising the variance of the learned data distribution and thereby degrading training stability and generative fidelity. Similarly, datasets of higher intrinsic complexity demand more iterations to capture fine-grained structural details. Consequently, greater complexity in either domain heightens sensitivity to the convergence rate. Under fixed complexity conditions, the convergence speed—governed by the eigenvalue—directly modulates the thoroughness of representation learning, ultimately determining generation quality.

## 4.4 QUANTIFYING THE EFFICIENCY-QUALITY TRADE-OFF

By Observation II and III above, under similar model-complexity and dataset conditions, the eigenvalue dominates both training speed and generative quality. Therefore, balancing these two factors could be achieved through eigenvalue modulation.

For linear $(a, b)$, we scale the coefficient of $b(x)$ in Tables 1 - 2, and report the corresponding results as in Tables 3. Eigenvalues in the range of approximately $(0.48, 5)$ on CIFAR-10 and $(0.48, 4)$ on CelebA-HQ-64 achieve an optimal balance between efficiency and quality relative to the baseline model, while the value $1.03$ both achieve this on Image-Net 128×128 and Image-Net 256×256 as in Tables 14 and 16.

However, for nonlinear $(a, b)$, the eigenvalue range may contract, as shown in Table 4. Moreover, with unknown $(a, b)$ or datasets, the range may still fluctuate. To avoid this uncertainty, we can adapt the scheduling function $\beta(t)$ according to the eigenvalue to achieve a balance. For example, when

Table 3: Comparison of optimal eigenvalue ranges for CIFAR-10 and CelebA-HQ-64

(a) CIFAR-10

| $a(x)$ | $b(x)$ | Eigenvalue | $T_{\text{conv}}$ | $D_{\text{disc}}$ | FID (SDE) | $T_{\text{spend}}$ |
|---|---|---|---|---|---|---|
| $\frac{1}{2}$ | -5$x$ | 6.15 | 525 | 0.209 | **4.30** | 26h |
| $\frac{1}{2}$ | -5.25$x$ | 6.40 | 512 | 0.209 | 4.48 | 27h |
| $\frac{1}{2}$ | -5.5$x$ | 6.65 | 500 | 0.209 | 4.63 | 26h |
| $\frac{1}{2}$ | -6$x$ | 7.20 | 475 | 0.209 | 4.97 | 26h |
| $\frac{1}{2}$ | -7$x$ | 8.30 | 425 | 0.209 | 5.47 | 26h |
| $\frac{1}{2}$ | -8$x$ | 9.40 | 375 | 0.209 | 5.92 | 24h |
| $\frac{1}{2}$ | -9$x$ | 10.50 | 325 | 0.209 | 6.28 | 21h |
| $\frac{1}{2}$ | -10$x$ | 10.70 | 350 | 0.208 | 6.33 | **20h** |

(b) CelebA-HQ-64

| $a(x)$ | $b(x)$ | Eigenvalue | $T_{\text{conv}}$ | $D_{\text{disc}}$ | FID (SDE) | $T_{\text{spend}}$ |
|---|---|---|---|---|---|---|
| $\frac{1}{2}$ | -2$x$ | 2.04 | 750 | 0.209 | **3.67** | 61h |
| $\frac{1}{2}$ | -3$x$ | 3.06 | 650 | 0.210 | 3.78 | 58h |
| $\frac{1}{2}$ | -4$x$ | 4.08 | 550 | 0.210 | 3.80 | 55h |
| $\frac{1}{2}$ | -4.5$x$ | 4.59 | 525 | 0.210 | 3.97 | 55h |
| $\frac{1}{2}$ | -4.75$x$ | 5.48 | 515 | 0.210 | 4.45 | 55h |
| $\frac{1}{2}$ | -5$x$ | 6.15 | 500 | 0.210 | 5.03 | **51h** |

$(a, b) = (\frac{1}{2}, -10x)$, the model has a large eigenvalue 10.7 but a high FID 6.33. Then taking $\frac{1}{10.7}\beta(t)$ as the new scheduling function, we get the balance as in Table 18. In fact, for a general $(a, b)$ needing fine-tuning, takeing the new scheduling function around (or marginally higher than) $\frac{0.48}{\lambda}\beta(t)$ is a recommended strategy, where 0.48 is the eigenvalue of the baseline.

This adjustment method works because, according to Theorem 2, speed also depends on $\beta(t)$. Crucially, $\beta(t)$ should be tuned with reference to the eigenvalue; otherwise the adjustment is blind. This demonstrates that controlling model efficiency and generation quality via the eigenvalue is feasible, and also shows that EGEA-DM is not a parameter-search model—even though some parameter tuning might sometimes be needed.

Figures 3-4 in Appendix C.1 show representative samples, confirming that spectral control preserves generation quality while accelerating diffusion.

### 4.5 COMPLEMENTARY STUDIES ON OTHER FACTORS

To guarantee adequate noise injection, the ($T_{conv}$) is finalized only after the $D_{\text{disc}}$ declines to a sufficiently low level and stabilizes. Empirical validation shows that further increasing the step count results in only marginal fluctuations in FID. A detailed analysis is provided in Appendix C.3, C.4 and Tables 9 - 12 there.

The final $D_{\text{disc}}$ differs among models as it relates to both the stationary and initial distributions by Theorem 2. Linear $(a, b)$ (Tables 1 - 3) exhibit more consistent distances owing to their relative simplicity and stability compared to nonlinear models (Tables 4 - 5 in Appendix C). From the experimental results, $D_{\text{disc}}$ does not significantly affect FID. See Appendix C.5 for more analysis.

Different $(a, b)$ typically correspond to distinct stationary distributions, yet changes in the stationary distribution have no significant impact on FID differences. See the Appendix C.6.

### 4.6 EGEA-DM AS A PLUG-AND-PLAY MODULE FOR DDPM ENHANCEMENTS

We evaluate EGEA-DM with classical ODE-based samplers, including DPM-Solver and DPM-Solver++ (Lu et al., 2022a;b). Tables 1 and 2 show that combining them with EGEA-DM can significantly achieve better generation quality. (Gray annotations correspond to DDPM.) Figures 1 and 2 and the corresponding Table 6 in Appendix C.1 illustrate the trend of FID with respect to the number of function evaluations (NFE), indicating that EGEA-DM outperforms the baseline DDPM and reflecting the stability of the our model.

These findings highlight the flexibility of EGEA-DM as a plug-in module for enhancing a wide range of diffusion model variants. Future work may explore adaptive eigenvalue scheduling to dynamically balance quality and efficiency.

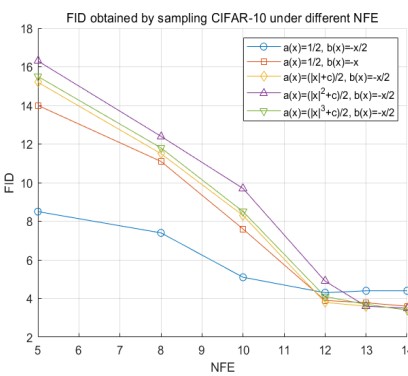
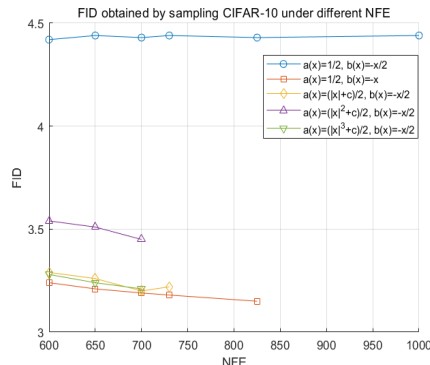

Figure 1: FID obtained by sampling CIFAR-10 under different NFE

Figure 2: FID obtained by sampling CIFAR-10 under different NFE

### 4.7 GENERALIZATION EXPERIMENT: DIFFERENT METHODS AND DATASETS

Based on the observations from the experimental results in the previous sections, we conducted generalization experiments on high-resolution datasets using SDE-based diffusion methods. The details are as follows:

The original DDPM model was trained on the ImageNet-128×128 dataset. Different sizes of linear operators were selected, and the appropriate $T_{\text{conv}}$ was determined with reference to the $D_{\text{disc}}$ of the baseline model. The results in the Table 14 indicate that the dominant eigenvalue governs the training time, and a slightly larger dominant eigenvalue leads to improved FID scores.

The EDMKarras et al. (2022) model was adopted for experiments on the CIFAR-10 dataset using the same parameters as those in the aforementioned table. The results in Table 15 consistently demonstrate the same trend as observed in the previous table.

The DiT(Diffusion Transformer)Peebles & Xie (2023) model was trained on the ImageNet-256×256 dataset with a fixed training duration, followed by multiple sampling processes. Table 16 show that the integration of EGEA achieves a lower FID score.

The DDIMSong et al. (2020a) method was used for sampling models under different operators to verify the robustness of EGEA - DM in the context of accelerated sampling. As indicated by the Table 17, EGEA - DM exhibits considerable stability.

### 4.8 EXPERIMENT SUMMARY

To summarize, this study conducts an in-depth investigation into the EGEA-DM model, with a particular focus on eigenvalue-related impacts and its plug-and-play capability. Experiments involving eigenvalue adjustment via $a(x)$ and $b(x)$ demonstrate that a larger principal eigenvalue generally leads to higher training efficiency, which is consistent with the conclusions derived from theoretical deductions. However, generative quality is affected by the coupling of multiple factors (e.g., the form of the diffusion operator, differences in dataset distributions, and the stationary distribution). Therefore, simultaneous improvements in both training speed and generative quality can be achieved by selecting an appropriate diffusion operator. Notably, when EGEA-DM is integrated with classical ODE-based samplers such as DPM-Solver and DPM-Solver++, its generative quality exhibits a significant improvement compared to DDPM, validating the model's flexibility as a plug-in. Furthermore, the generalization ability of EGEA-DM has been fully verified through integration with models including EDM, DiT and DDIM, as well as training and evaluation across a diverse range of datasets.

## 5 RELATED WORK

Diffusion Models (DMs) have demonstrated remarkable performance in generative tasks, yet their training process is plagued by critical limitations: substantial computational and memory overhead,

slow convergence rates, and the challenge of balancing generation quality with efficiency. These issues hinder their deployment in real-time applications.

To address these limitations, researchers have developed three key complementary categories of efficient training techniques. Firstly, latent diffusion maps data to low-dimensional latent spaces via autoencoders (AE), variational autoencoders (VAE). This approach enables a balance between generation quality and efficiency, as exemplified by models like LDMs(Rombach et al., 2022) and Stable Diffusion, which preserve high-quality generation while significantly improving training and inference efficiency. Secondly, loss function design is critical for DM efficiency and generation quality, such as CLDDockhorn et al. (2021) injects noise into data-coupled auxiliary variables to simplify learning. Thirdly, training tricks enhance DM efficiency, convergence, and quality. Such as DiGressVignac et al. (2022) optimizes efficiency for chemical molecules/social networks.

## 6 PRACTICAL GUIDELINES FOR USING EGEA-DM

Relative to nonlinear $(a, b)$, linear models exhibit superior stability and generation fidelity. Hence, we recommend the linear case as a preferred initialization. For either model class, appropriate $(a, b)$ can be selected by consulting the principles, empirical patterns and observations presented in Subsection 3.3 and Section 4. Additional fine-tuning may then be applied to attain a more desirable operating point on the speed–quality Pareto front.

When encountering novel $(a, b)$ configurations or unfamiliar datasets, if the initial performance (in either quality or speed) is suboptimal, our framework offers a principled two-step refinement protocol: (i) compute the principal eigenvalue via the analytical procedure outlined in Subsection 3.2 to locate the current operating regime; (ii) perform targeted adjustment using the $\beta(t)$-modulation method described in Section 4.4, which preserves theoretical guarantees while efficiently steering the model toward a satisfactory balance.

## 7 CONCLUSION

This paper proposes EGEA-DM, an eigenvalue-guided diffusion model framework that achieves principled acceleration and interpretability of diffusion models through spectral analysis of the $L$-generator. Leveraging ergodic theory, we relate the principal eigenvalue to convergence dynamics and introduce an adjustable mechanism to accelerate the training process without sacrificing generative quality. Extensive experiments across various datasets and models validate the effectiveness and efficiency of the framework. Adjusting the spectral properties of the diffusion generator shortens training time and significantly reduces the number of sampling steps while maintaining or improving generative quality (measured by the FID metric), demonstrating strong cross-architecture generalization ability. Furthermore, EGEA-DM can naturally integrate with existing methods such as DPM-Solver, exhibiting robust modularity.

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

## A  Theory of Time-Homogeneous Diffusion

Consider the diffusion process corresponding to $L$-generator in Eq. 4, the related diffusion equation of which is

$$dY_\tau = b(Y_\tau)d\tau + \sqrt{2a(Y_\tau)}dW_\tau, \tag{8}$$

The following theorem is about the uniqueness and ergodicity of the diffusion process related to $L$, which is given in (Chen & Mao, 2021, Section 7.4).

**Theorem 4.** *Given $Y_0$, the solution $Y_\tau$ of Eq. 8 is unique and ergodic if and only if*

$$\kappa(+\infty) = +\infty = \kappa(-\infty), \tag{9}$$

*and*

$$Z := \int_\mathbb{R} \frac{e^{C(u)}}{a(u)}du < +\infty, \tag{10}$$

*where*

$$\kappa(y) = \int_0^y e^{-C(z)}\left(\int_0^z \frac{e^{C(\xi)}}{a(\xi)}d\xi\right)dz, C(z) = \int_0^z \frac{b(\xi)}{a(\xi)}d\xi.$$

*If 10 holds, then the stationary distribution is*

$$\pi(dy) = \frac{1}{Za(y)}e^{C(y)}dy, \quad y \in \mathbb{R}. \tag{11}$$

The principal eigenvalue $\lambda_1$ in Eq. 6 has the variational formula as below. See (Chen, 2012, Theorem 3.2) for detail.

**Theorem 5.** *Under the condition 5, the variational formula of the principal eigenvalue $\lambda_1$ in Eq. 6 is*

$$\lambda_1 = \sup_{f \in \mathscr{C}_+} \left[ \inf_{z < \theta} II^-(f)(z)^{-1} \right] \bigwedge \left[ \inf_{z > \theta} II^+(f)(z)^{-1} \right].$$

*Here*

$$II^\pm(f) = \frac{h^\pm}{f}, \quad \mathscr{C}_+ = \{ f \in \mathscr{C}(-\infty, \infty) : f > 0 \},$$

*where*

$$h^-(z) = \int_{-\infty}^z e^{-C(x)} \, \mathrm{d}x \int_x^\theta \frac{e^C f}{a}, \quad z \leqslant \theta,$$

$$h^+(z) = \int_z^\infty e^{-C(x)} \, \mathrm{d}x \int_\theta^x \frac{e^C f}{a}, \quad z > \theta,$$

*and $\theta = \theta(f)$ is the unique root of the equation $h^-(\theta) = h^+(\theta)$.*

For any $B \in \mathcal{B}$, define $P_t^*(B) = \mathbb{P}\{Y_t \in B\}$. Denote the stationary distribution of $Y_t$ by $\pi^*(B) = \int_B \pi^*(dx)$. Then the convergence rate of $Y_t$ toward to $\pi^*$ is shown as below. See (Chen, 2005, Chapter 8).

**Theorem 6.** *Under the uniqueness and ergodicity conditions as in Theorem 4, it holds that*

$$\|P_t^* - \pi^*\|_{Var} \leq \|P_0^* - \pi^*\|_{Var} \, e^{-\lambda_1 t}.$$

## B  PROOFS

### B.1  PROOF OF THEOREM 1

*Proof.* Theorem 1 is derived directly from Theorem 4 and Lemma 1 and Lemma 2 below. $\square$

The following two lemmas show that Eq. 3 and 8 not only have the same uniqueness and ergodicity conditions, but have the same stationary distribution. The idea of proof is from (Bobrowski, 2008).

**Lemma 1.** *The following conditions are equivalent:*

    *(i) Given $X_0$, the solution $X_t$ to Eq. 3 is unique;*

    *(ii) Given $Y_0$, the solution $Y_\tau$ to Eq. 8 is unique;*

    *(iii) The boundary measure function $\kappa(y)$ satisfies 9.*

*Proof.* Define the time transformation

$$\phi(t) = \int_0^t \beta(s) \mathrm{d}s.$$

From equations Eq. 3 and Eq. 8, $X_t$ and $Y_\tau$ are related by

$$X_t = Y_{\phi(t)}, \quad Y_\tau = X_{\phi^{-1}(\tau)}.$$

Thus (i) and (ii) are equivalent. By Theorem 4, (ii) and (iii) are equivalent, which completes the equivalence of conditions (i)-(iii). $\square$

**Lemma 2.** *If the solutions to diffusion equations 3 and equation 8 are unique, then the following conditions are equivalent:*

    *(i) The solution $X_t$ to Eq.3 is ergodic;*

    *(ii) The solution $Y_\tau$ to Eq. equation 8 is ergodic;*

    *(iii) The normalization constant satisfies 5.*

*Proof.* By Lemma 1, $Y_\tau$ has uniqueness if and only if $\kappa(\pm\infty) = \infty$. According to Theorem 4, if $Y_\tau$ has uniqueness, then it is ergodic if and only if $Z < \infty$, with stationary distribution $\pi(y)$ in Eq. 11. Since $\tau(t)$ covers the entire time axis, for any Borel set $A \subset \mathbb{R}$,

$$\lim_{t \to \infty} \mathbb{P}(X_t \in A) = \lim_{\tau \to \infty} \mathbb{P}(Y_\tau \in A) = \pi(A).$$

Therefore, the limiting distribution of $X_t$ coincides with the stationary distribution of $Y_\tau$, and its ergodicity is equivalent to that of $Y_\tau$, proving the equivalence of conditions (i)-(iii). □

### B.2 PROOF OF THEOREM 2

*Proof.* Combine Theorem 6 and Lemma 1, we have

$$\|P_t - \pi\|_{\text{Var}} = \left\|P^*_{\phi(t)} - \pi^*\right\|_{\text{Var}} \le \|P^*_0 - \pi^*\|_{\text{Var}} \, e^{-\lambda_1 t}$$

$$= \|P_0 - \pi\|_{\text{Var}} \, e^{-\lambda_1 \int_0^t \beta(s)\mathrm{d}s}.$$

This completes the proof. □

### B.3 PROOF OF THEOREM 3

*Proof.* Suppose the proportion ality coefficient satisfies $\frac{b_2}{a_2} = \frac{b_1}{a_1}$ (i.e., $k = 1$), and the diffusion coefficient satisfies $a_2(x) = c \cdot a_1(x)$ where $c > 0$ is a constant.

Define the original operator:

$$L_1 = a_1(x)\partial_{xx} + b_1(x)\partial_x$$

and the scaled operator:

$$L_2 = a_2(x)\partial_{xx} + b_2(x)\partial_x = ca_1(x)\partial_{xx} + b_2(x)\partial_x$$

From the proportionality condition $\frac{b_2}{a_2} = \frac{b_1}{a_1}$, substituting $a_2 = ca_1$ gives $b_2(x) = c \cdot b_1(x)$. Thus:

$$L_2 = c\left[a_1(x)\partial_{xx} + b_1(x)\partial_x\right] = cL_1$$

meaning the scaled operator is a constant multiple of the original operator.

Let $L^*$ denote the adjoint operator of $L$. The stationary distribution $\pi_1$ of $L_1$ satisfies:

$$L_1^*\pi_1 = 0$$

The adjoint of the scaled operator satisfies $L_2^* = cL_1^*$, so:

$$L_2^*\pi_2 = 0 \iff cL_1^*\pi_2 = 0 \iff L_1^*\pi_2 = 0$$

Since the solution space of the adjoint equation $L_1^*\pi = 0$ is one-dimensional under normalization, we have $\pi_2 = \pi_1$, i.e., the stationary distributions are identical.

In the $L^2(\pi)$ space, for any function $f$ satisfying $\pi(f) = \int f\pi dx = 0$, the Dirichlet form of the scaled operator is:

$$\mathcal{E}_2(f, f) = \langle f, -L_2 f\rangle_\pi = \langle f, -cL_1 f\rangle_\pi = c\langle f, -L_1 f\rangle_\pi = c \cdot \mathcal{E}_1(f, f)$$

Here, the norm $\|f\|_\pi^2 = \int f^2\pi dx$ depends only on the stationary distribution $\pi$ and is independent of the operator coefficients.

The spectral gap is defined as the infimum of the Dirichlet form under the constraints $\pi(f) = 0$ and $\|f\|_\pi = 1$:

$$\lambda^{(2)} = \inf_{\substack{f:\pi(f)=0, \\ \|f\|_\pi=1}} \mathcal{E}_2(f, f)$$

Substituting the Dirichlet form relation yields:

$$\lambda^{(2)} = \inf_{\substack{f:\pi(f)=0, \\ \|f\|_\pi=1}} c \cdot \mathcal{E}_1(f, f) = c \cdot \inf_{\substack{f:\pi(f)=0, \\ \|f\|_\pi=1}} \mathcal{E}_1(f, f) = c \cdot \lambda^{(1)}$$

When the diffusion coefficient is scaled by a constant factor and the drift term maintains the same proportionality, the spectral gap is proportional to the scaling factor of the diffusion coefficient, while the stationary distribution remains unchanged. □

Table 4: Effect of $b(x)$ changes in nonlinear $L$-generators on CIFAR-10 and CelebA-HQ-64

(a) CIFAR-10

| $a(x)$ | $b(x)$ | Eigenvalue | $T_{\text{conv}}$ | $D_{\text{disc}}$ | FID↓ (SDE) | $T_{\text{spend}}$ |
|---|---|---|---|---|---|---|
| $\frac{|x|+c}{2}$ | $-0.4x$ | 0.42 | 800 | 0.175 | 3.45 | 45h |
| $\frac{|x|+c}{2}$ | $-0.5x$ | 0.52 | 730 | 0.192 | **3.22** | 41h |
| $\frac{|x|+c}{2}$ | $-0.6x$ | 0.62 | 680 | 0.210 | 3.85 | 38h |
| $\frac{|x|+c}{2}$ | $-x$ | 1.04 | 650 | 0.265 | 3.96 | 35h |
| $\frac{|x|+c}{2}$ | $-2x$ | 2.08 | 625 | 0.475 | 4.60 | **33h** |
| $\frac{x^2+c}{2}$ | $-0.5x$ | 0.66 | 700 | 0.320 | **3.45** | 43h |
| $\frac{x^2+c}{2}$ | $-0.65x$ | 0.86 | 650 | 0.355 | 4.12 | 40h |
| $\frac{x^2+c}{2}$ | $-x$ | 1.32 | 625 | 0.380 | 4.17 | 38h |
| $\frac{x^2+c}{2}$ | $-2x$ | 2.64 | 600 | 0.410 | 4.85 | **35h** |
| $\frac{|x|^3+c}{2}$ | $-0.35x$ | 1.08 | 700 | 0.375 | **3.21** | 42h |
| $\frac{|x|^3+c}{2}$ | $-0.5x$ | 1.54 | 625 | 0.410 | 3.35 | 38h |
| $\frac{|x|^3+c}{2}$ | $-0.75x$ | 2.31 | 550 | 0.475 | 4.60 | 33h |
| $\frac{|x|^3+c}{2}$ | $-x$ | 3.35 | 525 | 0.500 | 4.80 | 31h |
| $\frac{|x|^3+c}{2}$ | $-2x$ | 6.16 | 475 | 0.550 | 4.97 | **28h** |

(b) CelebA-HQ-64

| $a(x)$ | $b(x)$ | Eigenvalue | $T_{\text{conv}}$ | $D_{\text{disc}}$ | FID↓ (SDE) | $T_{\text{spend}}$ |
|---|---|---|---|---|---|---|
| $\frac{|x|+c}{2}$ | $-0.3x$ | 0.31 | 900 | 0.475 | 4.12 | 110h |
| $\frac{|x|+c}{2}$ | $-0.5x$ | 0.52 | 825 | 0.463 | **3.88** | 101h |
| $\frac{|x|+c}{2}$ | $-0.7x$ | 0.73 | 750 | 0.525 | 5.07 | 92h |
| $\frac{|x|+c}{2}$ | $-x$ | 1.04 | 725 | 0.550 | 5.60 | **88h** |
| $\frac{x^2+c}{2}$ | $-0.4x$ | 0.53 | 850 | 0.475 | 4.05 | 105h |
| $\frac{x^2+c}{2}$ | $-0.5x$ | 0.66 | 775 | 0.510 | **3.86** | 96h |
| $\frac{x^2+c}{2}$ | $-0.6x$ | 0.79 | 725 | 0.560 | 5.33 | 92h |
| $\frac{x^2+c}{2}$ | $-x$ | 1.32 | 700 | 0.575 | 5.77 | **84h** |

Table 5: $L$-generators with varying $a(x)$ and $b(x)$

| Datasets | $a(x)$ | $b(x)$ | Eigenvalue | $T_{\text{conv}}$ | $D_{\text{disc}}$ | FID (SDE solver) | $T_{spend}$ |
|---|---|---|---|---|---|---|---|
| CIFAR-10 | $0.5x^2 + 0.1$ | $-1.05x$ | 1.03 | 820 | 0.205 | 3.12 | 34h |
| CIFAR-10 | $0.3|x|^3 + 0.2$ | $-0.95x$ | 1.02 | 830 | 0.211 | 3.18 | 36h |
| CelebA-HQ-64 | $0.5x^2 + 0.1$ | $-1.05x$ | 1.03 | 795 | 0.495 | 5.05 | 89h |
| CelebA-HQ-64 | $0.3|x|^3 + 0.2$ | $-0.95x$ | 1.02 | 805 | 0.485 | 5.11 | 88h |

## C  ADDITIONAL EXPERIMENT DESCRIPTION

### C.1  ADDITIONAL EXPERIMENT RESULTS

Figures 3 and 4 show the image sampling results. Tables 6 present the specific values corresponding to Figures 1 and 2. Figure 5 illustrates the variation trend of the principal eigenvalue of the corresponding diffusion operator as the orders of $a(x)$ and $b(x)$ change. It can be observed that as the orders increase, the eigenvalues exhibit an upward trend. We explore the influence trend of ergodic theory on model performance under the scenario of nonlinear $L$-generator, and the results are presented in the Tables 4 and 5.

### C.2  VERIFICATION OF THE ERGODICITY AND UNIQUENESS OF THE DIFFUSION OPERATOR

See Table 8 and Table 7, we provide the verification results regarding the ergodicity and uniqueness of multiple diffusion operators for readers' reference, where ✓ denotes satisfaction and × denotes non-satisfaction.

### C.3  VERIFY THE IMPACT OF $T_{\text{CONV}}$ ON FID

Regarding the research on the impact on FID scores, to rule out the possibility that insufficient $T_{\text{conv}}$ were the cause, we additionally trained some operators in the Table 1 using the same $T_{\text{conv}}$. It can be seen that the FID scores showed almost no fluctuation due to the change in $T_{\text{conv}}$ in Table 9. This

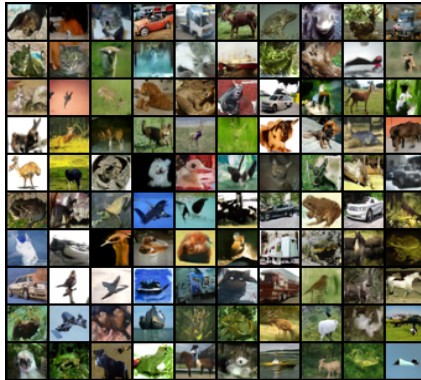

Figure 3: The generated image results
of EGEA-DM on CIFAR-10

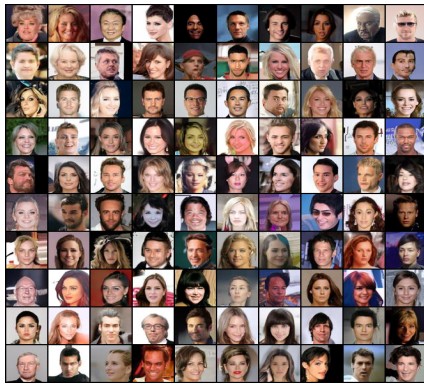

Figure 4: The generated image results
of EGEA-DM on CelebA

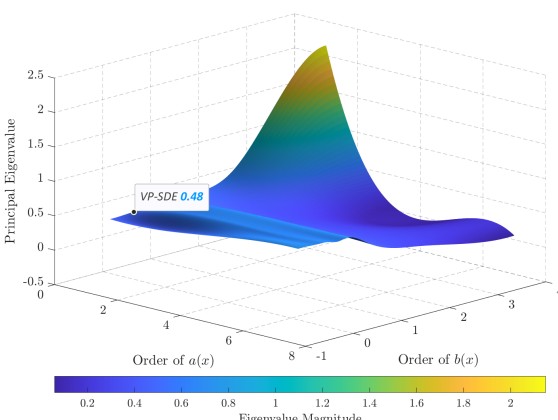

Figure 5: The principal eigenvalues of the generator depend on the orders of the functions $a$ and $b$, it shows the trend of changes in the principal eigenvalues as the order of $a$ and $b$ varies, which define its structure.

also indicates that compared with the baseline, the reduction in FID scores is not attributable to the decrease in $T_{\mathrm{conv}}$.

### C.4 ABOUT $D_{\mathrm{DISC}}$

In Tables 1-5, we explored the performance of various $a(x)$ and $b(x)$ when choosing a suitable $T_{\mathrm{conv}}$. In this section, we investigate how the model's performance changes when $a(x)$ and $b(x)$ are varied under a fixed $T_{\mathrm{conv}}$. As shown in the Table 12, compared with Table 4 (a), increasing $T_{\mathrm{conv}}$ does not lead to a decrease in $D_{\mathrm{disc}}$, which indicates that the distribution has converged. Meanwhile, the FID score does not fluctuate significantly and still shows the trend observed in Table 4 (a). This also suggests that the degradation of generative quality is not caused by insufficient sampling steps. Table 11 illustrates the variation trend of model performance when the $T_{\mathrm{conv}}$ in the forward noising process is insufficient. It can be observed that as the $T_{\mathrm{conv}}$ increases and the distribution approaches the stationary distribution more closely, the FID score exhibits a decreasing trend.

### C.5 ABOUT $D_0$

Based on the experimental results, it can be observed that under the linear condition, the $D_{\mathrm{disc}}$ consistently decreases as the eigenvalue increases, which aligns with our expectations. However, under the nonlinear condition, a distinct pattern emerges: despite the increase in eigenvalues, the

Table 6: FID obtained by sampling CIFAR-10

(a) FID obtained by sampling CIFAR-10 under different NFE

| $a(x)$ | $b(x)$ | solver | Eigenvalue | NFE=5 | NFE=8 | NFE=10 | NFE=12 | NFE=13 | NFE=14 |
|---|---|---|---|---|---|---|---|---|---|
| $\frac{1}{2}$ | $-0.5x$ | dpm solver | 0.48 | 8.50 | 7.40 | 5.10 | 4.30 | 4.40 | 4.40 |
| $\frac{1}{2}$ | $-x$ | dpm solver | 1.03 | 14.00 | 11.10 | 7.60 | 3.90 | 3.78 | 3.60 |
| $\frac{|x|+c}{2}$ | $-0.5x$ | sde solver | 0.52 | 15.20 | 11.50 | 8.30 | 3.80 | 3.60 | 3.50 |
| $\frac{x^2+c}{2}$ | $-0.5x$ | sde solver | 0.66 | 16.30 | 12.40 | 9.70 | 4.90 | 3.60 | 3.50 |
| $\frac{|x|^3+c}{2}$ | $-0.5x$ | sde solver | 1.54 | 15.50 | 11.80 | 8.50 | 4.10 | 3.70 | 3.40 |

(b) FID obtained by sampling CIFAR-10 under different NFE

| $a(x)$ | $b(x)$ | solver | NFE=600 | NFE=650 | NFE=700 | NFE=730 | NFE=825 | NFE=1000 |
|---|---|---|---|---|---|---|---|---|
| $\frac{1}{2}$ | $-0.5x$ | dpm solver | 4.42 | 4.44 | 4.43 | 4.44 | 4.43 | 4.44 |
| $\frac{1}{2}$ | $-x$ | dpm solver | 3.24 | 3.21 | 3.19 | 3.18 | 3.15 | - |
| $\frac{|x|+c}{2}$ | $-0.5x$ | sde solver | 3.29 | 3.26 | 3.20 | 3.22 | - | - |
| $\frac{x^2+c}{2}$ | $-0.5x$ | sde solver | 3.54 | 3.51 | 3.45 | - | - | - |
| $\frac{|x|^3+c}{2}$ | $-0.5x$ | sde solver | 3.28 | 3.24 | 3.21 | - | - | - |

Table 7: Cases where the diffusion operator satisfies ergodicity

| $a(x)$ \\ $b(x)$ | $-\frac{|x|^{-\frac{1}{2}}}{2}$ | $-\frac{|x|^{-\frac{1}{3}}}{2}$ | $-\frac{|x|^{-\frac{1}{20}}}{2}$ | $-\frac{|x|^{\frac{1}{20}}}{2}$ | $-\frac{|x|^{\frac{1}{3}}}{2}$ | $-\frac{|x|^{\frac{1}{2}}}{2}$ |
|---|---|---|---|---|---|---|
| $\frac{|x|^{-\frac{1}{2}}+c}{2}$ | × | × | × | × | × | × |
| $\frac{|x|^{-\frac{1}{3}}+c}{2}$ | × | × | × | × | × | × |
| $\frac{|x|^{-\frac{1}{20}}+c}{2}$ | × | × | × | × | × | × |
| $\frac{|x|^{\frac{1}{20}}+c}{2}$ | × | × | × | × | × | × |
| $\frac{|x|^{\frac{1}{3}}+c}{2}$ | × | × | × | × | × | × |
| $\frac{|x|^{\frac{1}{2}}+c}{2}$ | × | × | × | × | × | × |
| $\frac{|x|+c}{2}$ | × | × | × | × | × | × |
| $\frac{|x|^2+c}{2}$ | × | × | × | × | × | × |
| $\frac{|x|^3+c}{2}$ | × | × | × | × | × | × |
| $\frac{|x|^4+c}{2}$ | × | × | × | × | × | × |

$D_{\text{disc}}$ does not show a downward trend. Essentially, this phenomenon arises because as the diffusion operator changes, the stationary distribution also changes, and nonlinearity further enhances the diversity of the diffusion process.

To verify that the distance is continuously decreasing, we further analyzed the variation trend of distance across different steps. As showed in Table 10, the results confirm that this distance exhibits a decreasing tendency. Prior to this, we computed the corresponding $D_{\text{disc}}$ for different cases of $a(x)$ and $b(x)$, which corresponds to the left-hand side of Theorem 2. Due to the fact that the nonlinear $D_{\text{disc}}$ exhibits irregular magnitudes compared to the linear case, we proceed in this section by analyzing the right-hand side of the expression in the Theorem 2, hereinafter referred to as $D_0$. Based on the results in Table 13, $D_0$ tends to increase as $D_{\text{disc}}$ increases, which is consistent with the inequality stated in Theorem 2.

Table 8: Cases where the diffusion operator satisfies ergodicity

| $a(x)$ \ $b(x)$ | $-\frac{x}{2}$ | $-\frac{x^2}{2}$ | $-\frac{x^3}{2}$ | $-\frac{x^4}{2}$ | $-\frac{x^5}{2}$ | $-\frac{x^6}{2}$ | $-\frac{x^7}{2}$ |
|---|---|---|---|---|---|---|---|
| $\frac{|x|^{-\frac{1}{2}}+c}{2}$ | ✓ | × | ✓ | × | ✓ | × | ✓ |
| $\frac{|x|^{-\frac{1}{3}}+c}{2}$ | ✓ | × | ✓ | × | ✓ | × | ✓ |
| $\frac{|x|^{-\frac{1}{20}}+c}{2}$ | ✓ | × | ✓ | × | ✓ | × | ✓ |
| $\frac{|x|^{\frac{1}{20}}+c}{2}$ | ✓ | × | ✓ | × | ✓ | × | ✓ |
| $\frac{|x|^{\frac{1}{3}}+c}{2}$ | ✓ | × | ✓ | × | ✓ | × | ✓ |
| $\frac{|x|^{\frac{1}{2}}+c}{2}$ | ✓ | × | ✓ | × | ✓ | × | ✓ |
| $\frac{|x|+c}{2}$ | ✓ | × | ✓ | × | ✓ | × | ✓ |
| $\frac{|x|^2+c}{2}$ | ✓ | × | ✓ | × | ✓ | × | ✓ |
| $\frac{|x|^3+c}{2}$ | ✓ | × | ✓ | × | ✓ | × | ✓ |
| $\frac{|x|^4+c}{2}$ | ✓ | × | ✓ | × | ✓ | × | ✓ |

Table 9: $L$-generators with same steps on CIFAR-10

| $a(x)$ | $b(x)$ | Eigenvalue | $T_{\text{conv}}$ | FID | $T_{spend}$ |
|---|---|---|---|---|---|
| $\frac{1}{2}$ | $-0.25x$ | 0.24 | 1000 | 4.76 | 52h |
| $\frac{1}{2}$ | $-0.5x$ | 0.48 | 1000 | 4.44 | 45h |
| $\frac{1}{2}$ | $-x$ | 1.03 | 1000 | 3.15 | 40h |
| $\frac{1}{2}$ | $-2x$ | 2.04 | 1000 | 3.19 | 42h |
| $\frac{1}{2}$ | $-5x$ | 6.15 | 1000 | 4.27 | 44h |
| $\frac{1}{2}$ | $-10x$ | 10.70 | 1000 | 6.32 | 49h |
| $\frac{|x|+c}{2}$ | $-0.5x$ | 0.52 | 1000 | 3.22 | 50h |
| $\frac{|x|^2+c}{2}$ | $-0.5x$ | 0.66 | 1000 | 3.33 | 57h |
| $\frac{|x|^3+c}{2}$ | $-0.5x$ | 1.54 | 1000 | 3.18 | 63h |

## C.6 ABOUT $D_{\text{s}}$

To quantify the differences in stationary distributions corresponding to different operators, we define $D_{\text{s}}$ as the distance between the current stationary distribution and the baseline stationary distribution. As shown in Table 13, we calculated $D_{\text{s}}$ for a variety of diffusion operators, and the results indicate that $D_{\text{s}}$ varies with the operator. This not only reflects the variation trend of the stationary distribution as the operator changes but also indirectly demonstrates that the variation trend of FID is affected by the stationary distribution.

## D DISCRETE SGM

Given the forward SDE for the diffusion process as:

$$dx_t = \beta(t)b(x_t)dt + \sqrt{2\beta(t)}a(x_t)dW_t$$

where $W_t$ is a standard Brownian motion, $\beta(t)$ is the time-dependent diffusion coefficient, $b(x_t)$ and $a(x_t)$ are state-dependent drift/diffusion functions, $x_t \in \mathbb{R}^d$ denotes the state at time $t$.

The probability density $p_t(x)$ of $x_t$ satisfies the Fokker-Planck equation:

$$\partial_t p_t(x) = -\nabla_x \cdot (\beta(t)b(x)p_t(x)) + \frac{1}{2}\nabla_x^2 \cdot \left(2\beta(t)a(x)a^\top(x)p_t(x)\right)$$

Table 10: The $D_{\text{disc}}$ variation trend under different steps on CIFAR-10

| $a(x)$ | $b(x)$ | $T_{\text{conv}}$=625 | $T_{\text{conv}}$=650 | $T_{\text{conv}}$=680 | $T_{\text{conv}}$=730 | $T_{\text{conv}}$=800 |
|---|---|---|---|---|---|---|
| $\frac{|x|+c}{2}$ | $-0.4x$ | 1.030 | 0.915 | 0.725 | 0.515 | 0.175 |
| $\frac{|x|+c}{2}$ | $-0.5x$ | 0.805 | 0.695 | 0.485 | 0.192 | 0.195 |
| $\frac{|x|+c}{2}$ | $-0.6x$ | 0.605 | 0.460 | 0.210 | 0.215 | 0.210 |
| $\frac{|x|+c}{2}$ | $-x$ | 0.490 | 0.265 | 0.255 | 0.275 | 0.280 |
| $\frac{|x|+c}{2}$ | $-2x$ | 0.475 | 0.465 | 0.470 | 0.480 | 0.470 |

Table 11: FID variation trend under different $T_{\text{conv}}$ on CIFAR-10

| $a(x)$ | $b(x)$ | $T_{\text{conv}}$ | $D_{\text{disc}}$ | FID | $T_{\text{spend}}$ |
|---|---|---|---|---|---|
| $\frac{|x|+c}{2}$ | $-0.5x$ | 625 | 0.805 | 4.9 | 33h |
| $\frac{|x|+c}{2}$ | $-0.5x$ | 650 | 0.695 | 4.2 | 35h |
| $\frac{|x|+c}{2}$ | $-0.5x$ | 680 | 0.485 | 3.98 | 38h |

Reversing time $s = T - t$, the reverse SDE for $x_t$ (with $\tilde{W}_t$ as reverse Brownian motion) is:

$$dx_t = \left[ \beta(t)b(x_t) - 2\beta(t)a(x_t)a^\top(x_t)\nabla_x \log p_t(x_t) \right] dt + \sqrt{2\beta(t)}a(x_t)d\tilde{W}_t$$

where $\nabla_x \log p_t(x_t) = s_\theta(x_t, t)$ denotes the score function (modeled by $\theta$).

Discretize time into $0 = t_0 < t_1 < \cdots < t_N = T$, with $\beta_i = \beta(t_i)$, $b_i = b(x_{t_i})$, $a_i = a(x_{t_i})$.

$$x_{i+1} = x_i + \beta_i b(x_i)\Delta t + \sqrt{2\beta_i \Delta t}a(x_i)z_i, \quad z_i \sim \mathcal{N}(0, \boldsymbol{I})$$

Using $s_\theta(x_{i+1}, i+1) = \nabla_x \log p_{i+1}(x_{i+1})$, the reverse iteration is:

$$x_i = x_{i+1} - \beta_{i+1}b(x_{i+1})\Delta t + 2\beta_{i+1}a(x_{i+1})a^\top(x_{i+1})s_\theta(x_{i+1}, i+1)\Delta t + \sqrt{2\beta_{i+1}\Delta t}a(x_{i+1})\tilde{z}_{i+1}$$

where $\tilde{z}_{i+1} \sim \mathcal{N}(0, \boldsymbol{I})$ and $i = 0, 1, \ldots, N-1$.

Absorbing $\Delta t$ into coefficients (simplified notation):

$$x_i = x_{i+1} - \beta_{i+1}b(x_{i+1}) + 2\beta_{i+1}a(x_{i+1})a^\top(x_{i+1})s_\theta(x_{i+1}, i+1) + \sqrt{2\beta_{i+1}}a(x_{i+1})\tilde{z}_{i+1}$$

# E  SUPPLEMENTARY EXPERIMENTS

Table 12: Study the influence of different $a(x)$ and $b(x)$ on FID with a fixed $T_{\text{conv}}$ on CIFAR-10

| $a(x)$ | $b(x)$ | Eigenvalue | $T_{\text{conv}}$ | $D_{\text{disc}}$ | FID↓ (SDE) | $T_{\text{spend}}$ |
|---|---|---|---|---|---|---|
| $\frac{|x|+c}{2}$ | $-0.4x$ | 0.42 | 800 | 0.175 | 3.45 | 45h |
| $\frac{|x|+c}{2}$ | $-0.5x$ | 0.52 | 800 | 0.195 | **3.12** | 45h |
| $\frac{|x|+c}{2}$ | $-0.6x$ | 0.62 | 800 | 0.210 | 3.63 | 45h |
| $\frac{|x|+c}{2}$ | $-x$ | 1.04 | 800 | 0.280 | 3.68 | 45h |
| $\frac{|x|+c}{2}$ | $-2x$ | 2.08 | 800 | 0.470 | 4.24 | **45h** |

Table 13: Study the Dis of different $a(x)$ and $b(x)$ on CIFAR-10

| $a(x)$ | $b(x)$ | Eigenvalue | $T_{\text{conv}}$ | $D_{\text{disc}}$ | $D_0$ | $D_s$ | FID |
|---|---|---|---|---|---|---|---|
| $\frac{1}{2}$ | $-0.25x$ | 0.24 | 1000 | 0.241 | 27.3 | 2.1 | 4.76 |
| $\frac{1}{2}$ | $-0.5x$ | 0.48 | 1000 | 0.208 | 25.8 | 0 | 4.44 |
| $\frac{1}{2}$ | $-x$ | 1.03 | 825 | 0.209 | 26.7 | 1.3 | 3.15 |
| $\frac{1}{2}$ | $-2x$ | 2.04 | 750 | 0.208 | 28.2 | 3.4 | 3.19 |
| $\frac{1}{2}$ | $-5x$ | 6.15 | 525 | 0.209 | 30.6 | 6.5 | 4.30 |
| $\frac{1}{2}$ | $-10x$ | 10.70 | 350 | 0.208 | 34.8 | 12.1 | 6.33 |
| $\frac{|x|+c}{2}$ | $-0.4x$ | 0.42 | 800 | 0.175 | 34.5 | 11.5 | 3.45 |
| $\frac{|x|+c}{2}$ | $-0.5x$ | 0.52 | 730 | 0.192 | 30.4 | 6.4 | 3.22 |
| $\frac{|x|+c}{2}$ | $-0.6x$ | 0.62 | 680 | 0.210 | 26.5 | 1.1 | 3.85 |
| $\frac{|x|+c}{2}$ | $-x$ | 1.04 | 650 | 0.265 | 27.8 | 2.7 | 3.96 |
| $\frac{|x|+c}{2}$ | $-2x$ | 2.08 | 625 | 0.475 | 32.5 | 9.4 | 4.60 |
| $\frac{x^2+c}{2}$ | $-0.5x$ | 0.66 | 700 | 0.320 | 28.4 | 3.6 | 3.45 |
| $\frac{|x|^3+c}{2}$ | $-0.5x$ | 1.54 | 700 | 0.375 | 31.2 | 7.5 | 3.21 |
| $0.5x^2 + 0.1$ | $-1.05x$ | 1.03 | 820 | 0.205 | 26.9 | 1.5 | 3.12 |
| $0.3|x|^3 + 0.2$ | $-0.95x$ | 1.02 | 830 | 0.211 | 27.1 | 1.8 | 3.18 |

Table 14: $L$-generators with fewer steps on Image-Net 128×128

| $a(x)$ | $b(x)$ | Eigenvalue | $T_{\text{conv}}$ | $D_{\text{disc}}$ | FID | $T_{spend}$ |
|---|---|---|---|---|---|---|
| $\frac{1}{2}$ | $-0.5x$ | 0.48 | 1000 | 0.763 | 35.24 | 182h |
| $\frac{1}{2}$ | $-x$ | 1.03 | 800 | 0.765 | **31.63** | 164h |
| $\frac{1}{2}$ | $-10x$ | 10.70 | 325 | 0.761 | 42.58 | **92h** |

Table 15: $L$-generators with fewer steps on Cifar-10 with EDM

| $a(x)$ | $b(x)$ | Eigenvalue | $T_{\text{conv}}$ | FID | $T_{spend}$ |
|---|---|---|---|---|---|
| $\frac{1}{2}$ | $-0.5x$ | 0.48 | 1000 | 13.22 | 120h |
| $\frac{1}{2}$ | $-x$ | 1.03 | 825 | **11.27** | 101h |
| $\frac{1}{2}$ | $-10x$ | 10.70 | 350 | 18.48 | **67h** |

Table 16: $L$-generators with fewer steps with DiT on Image-Net 256×256

| $a(x)$ | $b(x)$ | Eigenvalue | $T_{\text{conv}}$ | FID1 | FID2 | FID3 |
|---|---|---|---|---|---|---|
| $\frac{1}{2}$ | $-0.5x$ | 0.48 | 1000 | 270 | 220 | 198 |
| $\frac{1}{2}$ | $-x$ | 1.03 | 825 | 243 | 200 | 182 |

Table 17: $L$-generators with fewer steps on Cifar-10 with DDIM

| $a(x)$ | $b(x)$ | Eigenvalue | $T_{\text{conv}}$ | $D_{\text{disc}}$ | FID | $T_{spend}$ |
|---|---|---|---|---|---|---|
| $\frac{1}{2}$ | -0.25x | 0.24 | 1000 | 0.241 | 5.08 | 52h |
| $\frac{1}{2}$ | -0.5x | 0.48 | 1000 | 0.208 | 4.72 | 45h |
| $\frac{1}{2}$ | -x | 1.03 | 825 | 0.209 | **3.38** | 35h |
| $\frac{1}{2}$ | -10x | 10.70 | 350 | 0.208 | 6.57 | **20h** |

Table 18: By modifying $\beta(t)$ to correct the excessively fast speed

| $a(x)$ | $b(x)$ | Eigenvalue | Scheduling Function | $T_{\text{conv}}$ | $D_{\text{disc}}$ | FID | $T_{spend}$ |
|---|---|---|---|---|---|---|---|
| $\frac{1}{2}$ | -10x | 10.70 | $\beta(t)$ | 350 | 0.208 | 6.33 | 20h |
| $\frac{1}{2}$ | -10x | 10.70 | $\frac{1.03}{10.70}\beta(t)$ | 700 | 0.208 | 3.88 | 28h |

