# OpenReview forum: "EGEA-DM: Eigenvalue-Guided Explainable and Accelerated Diffusion Model"
_ICLR.cc/2026/Conference — ICLR 2026 Conference Desk Rejected Submission_

### Official Review · Reviewer_TSKY · 2025-10-28

**Soundness:** 2
**Presentation:** 2
**Contribution:** 2
**Rating:** 4
**Confidence:** 2

**Summary:**

This paper aims to enhance the training efficiency, interpretability, and controllability of diffusion models. Drawing on ergodic theory, the authors proposed an Eigenvalue-Guided Explainable and Accelerated Diffusion Model (EGEA-DM). This approach leverages the principal eigenvalue of the L-generator to precisely regulate the forward diffusion speed, thereby enabling adaptive adjustment of reverse steps during both training and sampling phases.

**Strengths:**

1、Reinterpreting diffusion models through the lens of ergodic theory, while establishing a connection between the convergence rate of the forward process and the principal eigenvalue of the L-generator, is an interesting idea.
2、By tuning the coefficients of the L-generator according to its principal eigenvalue, the proposed method introduces a flexible mechanism that enables control over both the speed and stability of the diffusion process.

**Weaknesses:**

1、It appears that as the eigenvalue increases, the FID of the model's generated results may deteriorate. Given that the appropriate eigenvalue range is critical for balancing efficiency and generative performance, how should this range be determined? Is it necessary to conduct careful parameter tuning for different models individually?
2、Both qualitatively and quantitatively, the paper lacks comparisons between the proposed method, a baseline (i.e., the model without using the proposed method), and other SOTA approaches. Consequently, the true performance gains brought by the proposed method cannot be accurately evaluated. Furthermore, experiments are only conducted on two small-scale or low-resolution datasets—CIFAR-10 and CelebA-HQ-64, with no experimental validation of generalization on larger-scale or higher-resolution datasets.
3、The latest methods discussed in the "Related Work" section only date up to 2023, lacking discussions on the connections to and distinctions from more recent advances.

**Questions:**

Please see weaknesses above.

---

> ### Author Response · Authors · 2025-11-21
> **Response to Reviewer TSKY**
>
> Thank you for your review comments on our paper and your recognition of certain aspects. Regarding the weaknesses and questions you mentioned, we provide explanations below and have made corresponding revisions to the paper (see the revised version). We will also submit an updated version by December 3nd, which includes supplementary experiments.
>
> If you have any questions about our response or if we have misunderstood your comments, please feel free to raise further questions or clarify your concerns.
>
> **Weakness/Question 1.  For “balancing efficiency and generative performance”**
>
> Indeed, the experimental results demonstrate that the generation quality degrades when the eigenvalue increase beyond a certain threshold. This phenomenon may be attributed to the excessively rapid evolution rate, which impairs training stability and data diversity. Identifying the eigenvalue range—even the optimal balance point—holds significant research value. However, theoretically determining this range remains an open and highly challenging problem. Practically, it is also complex to pinpoint, as experimental results indicate that the range is influenced by numerous factors, such as the dataset, the selection of $(a,b)$, and the sampler.
>
> Nevertheless, our experimental findings and observations provide valuable references for researchers seeking to identify this range. Regarding the evolution rate (i.e., the magnitude of eigenvalues), readers can directly refer to our computed values and their variation trends with respect to $(a, b)$. This is because the eigenvalues change continuously with $(a, b)$ under certain conditions, as corroborated by the Kato-Rellich theorem.
>
> For generation quality, multiple factors come into play. It is foreseeable that the model architecture and complexity will affect the eigenvalue range. Nevertheless, based on the experimental patterns we have identified, readers can adjust the eigenvalues and select appropriate $(a, b)$ pairs to achieve optimization goals.
>
> In future work, we will attempt to conduct further exploration from both theoretical and experimental perspectives.
>
> **Weakness/Question 2. For "Generalization validation"**
>
> Your question is highly pertinent. In fact, we have also been considering these two aspects.
>
> Currently, there exists a wealth of research based on classical diffusion models. However, most of these research methods exhibit strong dependence on the values and properties of the baseline $(a, b)$ parameters. In contrast, this work considers different configurations of $(a, b)$ that no longer preserve these properties. Consequently, existing methods cannot be directly adapted to our framework and require further in-depth investigation, which constitutes a key component of our future work.
>
> For methods that are potentially adaptable, we are conducting relevant experiments. We will include these experimental results in the updated version of the paper to be submitted on December 3nd.
>
> Regarding experiments on large-scale or higher-resolution datasets, we are currently conducting tests. These results will also be incorporated into the updated paper submitted on December 3nd.
>
> **Weakness/Question 3.  For "Related Work"**
>
> We will refine the section the "Related Work" section in light of your comments.
>
>
> **Revision Statement**
>
> In the paper, we have revised the potentially ambiguous sections and emphasized the core research contributions. To better illustrate our descriptions, the updated version  includes the following additional experiments.
> + Ergodicity verification of operators with different forms.
> + Model performance under synchronous training steps, verifying the impact of sampling steps on performance.
> + Experiments on high-resolution datasets and more Diffusion models (before December 3nd).

---

> ### Author Response · Authors · 2025-12-03
> **Revision Notes**
>
> Dear Reviewers,
>
> We sincerely appreciate your valuable comments and professional suggestions on our paper titled *EGEA-DM: Eigenvalue-Guided Explainable and Accelerated Diffusion Model*. Your meticulous reviews have provided crucial guidance for us to identify research limitations and enhance the quality of this work. We are deeply grateful for your efforts and have carefully addressed each of your comments with corresponding revisions. The detailed modifications are outlined as follows:
>
> ## Key Revisions
> 1. We have elaborated on the application of the one-dimensional theory to multi-dimensional data, clarifying the rationale behind the validity of extending the one-dimensional theoretical framework to multi-dimensional datasets for readers' better understanding.
> 2. To verify the generalization ability of our method, we have expanded the experimental evaluations by incorporating additional model architectures and high-resolution datasets.
> 3. Relevant content regarding eigenvalue calculation has been supplemented to provide readers with a more comprehensive understanding of our proposed approach.
> 4. Inappropriate descriptions throughout the paper have been revised to avoid any potential misunderstandings.
>
> It should be noted that the revised parts in the paper are highlighted in blue font.
>
> We would like to express our sincere gratitude again for your thoughtful reviews. All revisions have been completed to thoroughly address your comments and suggestions. Please feel free to inform us if further improvements are needed, and we will continue to refine the paper accordingly.
>
> Sincerely,
> The Authors of *EGEA-DM: Eigenvalue-Guided Explainable and Accelerated Diffusion Model*.

---

### Official Review · Reviewer_sWwS · 2025-10-30

**Soundness:** 3
**Presentation:** 4
**Contribution:** 2
**Rating:** 4
**Confidence:** 4

**Summary:**

This paper introduces EGEA-DM, a framework intended to address the high computational costs, slow convergence, and perceived lack of theoretical interpretability in existing diffusion models. The central thesis is the application of ergodic theory to the generative diffusion process. Specifically, the authors propose a formal connection between the convergence rate of the forward diffusion process and the principal eigenvalue ($\lambda_1$) of its corresponding L-generator. The framework's core mechanism involves modulating this principal eigenvalue by adjusting the coefficients $a(x)$ and $b(x)$ of the L-generator. The authors provide a theoretical argument (Theorem 2) and empirical results (on CIFAR-10 and CelebA-HQ-64) to support their main claim: a larger principal eigenvalue $\lambda_1$ leads to an exponentially faster convergence to the stationary distribution. This, in turn, allows the model to reach convergence in fewer forward steps, significantly reducing the computational overhead for both training and sampling. The method is also presented as a plug-and-play module compatible with existing samplers like DPM-Solver.

**Strengths:**

- The primary strength of this work is its effort to ground the problem of diffusion model acceleration in established mathematical theory. While the underlying concepts of spectral gap convergence for 1D diffusions (e.g., from Chen 2005/2012, Bobrowski 2008) are not new, their application as a design principle for accelerating generative models is a valuable contribution.
- The paper does not just posit a theoretical relationship. It provides a operational method for implementing its core idea by adapting an iterative numerical estimation algorithm (from Chen 2012) to estimate the principal eigenvalue $\lambda_1$. This turns a purely theoretical quantity into an actionable component of the algorithm.
- The paper is generally well-written and logically structured. The narrative guides the reader from the ergodic theory, through the specifics of the L-generator and its eigenvalue, to the eventual experimental validation.

**Weaknesses:**

- The entire theoretical derivation (Preliminaries, Section 3.1) is explicitly simplified to a one-dimensional (1D) case for tractability. However, diffusion models for image generation operate in extremely high-dimensional spaces. The paper makes no serious attempt to bridge this enormous theoretical gap. Spectral gap analysis in high dimensions is notoriously more complex than in 1D and often requires additional structural assumptions. The authors do not discuss how, or even if, their 1D-derived insights generalize to high-dimensional, non-reversible, or anisotropic processes common in diffusion modeling.
- The proofs for the main theorems, such as Theorem 2, are relegated to the appendix and are overly brief. They lack the rigor and detail necessary for a reader to verify the claims or, more importantly, to understand the precise preconditions and scope under which these convergence bounds hold. This ambiguity undermines the paper's theoretical foundation.
- The paper claims to provide an "explainable" framework. However, "Guiding Principle II" (which states that higher polynomial orders for $a(x)$ and $b(x)$ lead to a larger $\lambda_1$) is presented as a purely empirical observation from numerical experiments. This is a heuristic, not an explanation. The paper fails to provide any theoretical justification for why this relationship holds, which runs counter to its own stated objective of improving interpretability.

Collectively, these issues suggest that the theoretical contribution of the paper is largely limited to restating classical diffusion results in a simplified 1D setting, without sufficient rigor or justification to claim novel theoretical insight.

**Questions:**

The proposed method introduces an extra, non-trivial computational step: the iterative numerical estimation of $\lambda_1$. The appendix (C.2) claims this overhead is "entirely acceptable." However, the framework's performance seems to hinge critically on this value. How sensitive is the model's performance (e.g., final FID, optimal $T_{conv}$) to the accuracy of this numerical estimation? If the estimation error is, for example, 10% or 20%, does the principled acceleration break down or become suboptimal?

---

> ### Author Response · Authors · 2025-11-21
> **Response to Reviewer sWwS (Part 1)**
>
> Thank you for your review comments on our paper and your recognition of certain aspects. Regarding the weaknesses and questions you mentioned, we provide explanations below and have made corresponding revisions to the paper (see the revised version). We will also submit an updated version by December 3nd, which includes supplementary experiments.
> If you have any questions about our response or if we have misunderstood your comments, please feel free to raise further questions or clarify your concerns.
>
> **Response for Weaknesses**
>
> **W1. For “The theory lives in 1D but the experiments are in high dimensions”**
>
> Indeed, the data used in the experiments is extremely high-dimensional and contains structural information across dimensions. However, all the theories presented in the paper pertain to one-dimensional diffusion processes. This discrepancy may indeed raise questions among readers. We argue that extending the one-dimensional theory to high-dimensional scenarios is feasible, with the following justifications.
>
> Structural information across dimensions is embedded in the data distribution but does not interfere with the noising and denoising processes, thus having no impact on the convergence rate. In fact, each dimension undergoes noising and denoising independently following the same SDEs, which corresponds to the identical $L$-operator, stationary distribution, and the principle eigenvalue—only the initial distribution of each dimension may vary. Therefore, the convergence rate of high-dimensional data can be characterized by these shared eigenvalue.
>
> This paper focuses on accelerating diffusion models based on the eigenvalue while ensuring generation quality through FID,  a two-fold approach which is not  affected by the structural information across dimensions. Experimental results also validate the effectiveness of our strategy.
>
> The aforementioned analysis has been incorporated into the revised version of the paper, replacing the concise statement in the original draft.
>
> **W2. For "Succinct theoretical description”**
>
> We will revise the proof of Theorem 2 to enhance its readability for readers in the version submitted on December 2nd.
>
> **W3. For "Inappropriate description”**
>
> We apologize for the insufficient clarity and comprehensiveness in the original manuscript, which may have led to misunderstandings regarding the paper’s framework. The objectives of this work are twofold: firstly, to accelerate diffusion models based on the ergodic theory of diffusion processes while ensuring generation quality through observations; secondly, to explore alternative forms beyond the classical $(a, b)$ framework, thereby paving the way for expanding diffusion model research.
>
> We conducted experiments using polynomial functions for two purposes: on the one hand, to verify the feasibility of acceleration guided by the eigenvalue and its consistency with the proposed theory; on the other hand, to evaluate the generation quality, which is empirical rather than theoretically grounded. These observations and empirical findings can provide readers with practical references for model parameter selection.
>
> The comment that “Guiding principle II is a heuristic, not an explanation” is entirely valid. The term “Guiding principle” was an inappropriate choice in the original manuscript. In the revised version, we have replaced it with “Observation”. Additionally, we have refined the model description in the new draft to facilitate readers’ understanding of our proposed framework.

---

> ### Author Response · Authors · 2025-11-21
> **Response to Reviewer sWwS (Part 2)**
>
> **Response to Questions**
>
> 1). _For " accuracy of numerical estimation"._ The purpose of calculating eigenvalues is to verify the consistency between theory and experiments by comparing their magnitudes. While higher computational precision is desirable, it is not the primary objective of this study.
>
> Nevertheless, your concern is valid. Errors in eigenvalue estimation must not interfere with the comparison of eigenvalues or the evaluation of model performance. Therefore, we adopt Chen’s estimation method, as it is theoretically exact and computationally tractable for numerical implementation. Typically, the true values of eigenvalues are difficult to derive through mathematical deductions, except for certain special diffusion processes. For derivable cases (e.g., DDPM or other corresponding diffusion operators), we have evaluated the estimation errors, which are approximately 0.007.
>
> 2). _For " computational time"._ It is important to supplement that the eigenvalue calculation was performed on an Intel Core i5-9300H processor using MATLAB, while model training and generation were conducted on an NVIDIA GeForce RTX 4090 GPU. Therefore, if the eigenvalue calculation is executed on hardware with higher specifications, the computational time is expected to be significantly reduced.
>
> 3). _For "the claim"._ Additionally, we apologize for claiming that "the computational cost of over two hours is entirely acceptable" in the original manuscript—this was an inappropriate formulation. In the revised version, we have corrected this statement and supplemented additional explanations regarding the eigenvalue calculation.
>
> **Revision Statement**
>
> In the paper, we have revised the potentially ambiguous sections and emphasized the core research contributions. To better illustrate our descriptions, the updated version  includes the following additional experiments.
> + Ergodicity verification of operators with different forms.
> + Model performance under synchronous training steps, verifying the impact of sampling steps on performance.
> + Experiments on high-resolution datasets and more Diffusion models (before December 3nd).

---

> > ### Comment · Reviewer_sWwS · 2025-11-26
> > **Response to authors.**
> >
> > Thank you for the revisions regarding the computational time and related clarifications, and I also appreciate the additional experiments. However, I still have several concerns. First, I am grateful for the expanded proof of Theorem 2, but after reading the relevant arguments in Appendix A, it appears that the theoretical components rely almost entirely on prior work (Chen & Mao, 2021) or standard textbook results, with little technical innovation. In addition, as several other reviewers have also pointed out, I remain unconvinced about the interpretability of the (a,b)-framework. Finally, in the authors’ response to W1, the claim that FID is “not affected by the structural information across dimensions” seems questionable to me. Since the definition of FID explicitly involves the covariance matrix, how can it be unaffected by structural information?

---

> ### Author Response · Authors · 2025-12-03
> **Response to Reviewer sWwS**
>
> Thank you for your reply to our response. Following your comments, we provide explanations below and have made corresponding revisions to the manuscript (see the revised version).
>
> ### **1. For the theory**
> The paper indeed utilizes the ergodic theory of time-homogeneous diffusion processes from [Chen&Mao, 2021], but this theory cannot be directly applied to diffusion models. The main theoretical contributions of the paper lie in proposing a new theoretical framework suitable for diffusion models based on this existing theory, and further providing a novel theoretical perspective for optimizing such models. Specifically:
>
> (1) Using the conclusions from [Chen&Mao, 2021], we derive the ergodic theory for time-inhomogeneous diffusion processes, including the equivalent conditions for uniqueness and ergodicity (Theorem 1), as well as the characterization of convergence rate (Theorem 2). This forms the theoretical foundation of the paper.
>
> (2) Based on the theory in (1), the paper proposes a new research framework for diffusion models (see the SDE and L operator in the manuscript). Theoretically, it not only extends the classic DDPM framework but also deeply explores the SGM framework. In fact, experimentally, the paper also extends the commonly adopted form (a=1, b=-1/(2x)).
>
> (3) Building on (1) and (2), the paper proposes a new perspective for model acceleration with reference to eigenvalues.
>
> In the revised manuscript, we have elaborated on the theoretical contributions and improved the theoretical derivation process.
>
> ### **2. For the interpretability of the (a,b)-framework**
> Our model achieves acceleration while improving or maintaining generation quality with reference to eigenvalues. This is based on the ergodic theory of diffusion processes and universal laws, making it interpretable. The detailed analysis is as follows:
>
> (1) The theory provides a clear search space and interpretable regulation mechanism
> Our framework is based on ergodic theory, with its core being the use of eigenvalues to quantify the speed of the diffusion process. Theoretical analysis demonstrates an explicit monotonic relationship between eigenvalues and training speed. This transforms hyperparameter search from an unbounded, unguided process into one with tangible references. Adjusting parameters (a, b) essentially enables continuous, predictable regulation of "speed" under theoretical guarantees, rather than blind "trial-and-error".
>
> (2)  Experiments aim to verify the theory and reveal universal laws, not to perform parameter search
> The primary purpose of our extensive experiments is to validate the impact of eigenvalues on speed and generation quality. Secondly, we aim to uncover how two factors—model complexity and dataset complexity—interact with "speed" to collectively influence the final generation quality.
>
> Experiments show that under fixed model and dataset conditions, there exists a clear increasing-then-decreasing trend between generation quality (FID) and eigenvalues (speed). This confirms the predictability and interpretability of our model. The universal laws revealed by the experiments (e.g., linear models are generally more stable) provide users with direct, experience-based priors, which further reduce rather than increase future parameter tuning costs.
> ### **3. For authors’ response to W1**
> Thanks for prompting us to clarify this point. Our response regarding W1 aimed to convey that structural information across dimensions affects neither (1) accelerating diffusion models through eigenvalue analysis, nor (2) ensuring generation quality via FID evaluation.
>
> You are correct that the computation of FID can indeed be affected by structural information. We do not dispute this point.
>
> We have revised our response to W1 to eliminate the ambiguity.
>
> We sincerely thank you for your valuable time and comments, which have greatly improved the quality of our work. We hope the above responses and manuscript revisions address your remaining concerns.

---

> ### Author Response · Authors · 2025-12-03
> **Revision Notes**
>
> Dear Reviewers,
>
> We sincerely appreciate your valuable comments and professional suggestions on our paper titled *EGEA-DM: Eigenvalue-Guided Explainable and Accelerated Diffusion Model*. Your meticulous reviews have provided crucial guidance for us to identify research limitations and enhance the quality of this work. We are deeply grateful for your efforts and have carefully addressed each of your comments with corresponding revisions. The detailed modifications are outlined as follows:
>
> ## Key Revisions
> 1. We have elaborated on the application of the one-dimensional theory to multi-dimensional data, clarifying the rationale behind the validity of extending the one-dimensional theoretical framework to multi-dimensional datasets for readers' better understanding.
> 2. To verify the generalization ability of our method, we have expanded the experimental evaluations by incorporating additional model architectures and high-resolution datasets.
> 3. Relevant content regarding eigenvalue calculation has been supplemented to provide readers with a more comprehensive understanding of our proposed approach.
> 4. Inappropriate descriptions throughout the paper have been revised to avoid any potential misunderstandings.
>
> It should be noted that the revised parts in the paper are highlighted in blue font.
>
> We would like to express our sincere gratitude again for your thoughtful reviews. All revisions have been completed to thoroughly address your comments and suggestions. Please feel free to inform us if further improvements are needed, and we will continue to refine the paper accordingly.
>
> Sincerely,
> The Authors of *EGEA-DM: Eigenvalue-Guided Explainable and Accelerated Diffusion Model*.

---

### Official Review · Reviewer_MbUR · 2025-10-31

**Soundness:** 2
**Presentation:** 1
**Contribution:** 1
**Rating:** 4
**Confidence:** 4

**Summary:**

The paper proposes EGEA-DM, which applies ergodic theory to diffusion models by controlling the principal eigenvalue of the L-generator to regulate forward diffusion speed. The authors claim this provides theoretical interpretability and enables faster training while maintaining generation quality. Experiments on CIFAR-10 and CelebA-HQ-64 show reduced training steps, though with mixed results on quality.

**Strengths:**

1. **Interesting theoretical angle.** Connecting the spectral gap of the L-generator to convergence rates is a neat idea, and framing diffusion acceleration through ergodic theory is relatively unexplored in the generative modeling literature.

2. **Rigorous mathematical development (in 1D).** Theorems 1-2 properly characterize uniqueness, ergodicity, and convergence rates for one-dimensional diffusion processes. The eigenvalue estimation procedure based on Chen (2012) is technically sound.

3. **Integration with existing samplers.** The framework works with DPM-Solver and DPM-Solver++, suggesting it could be modular.

**Weaknesses:**

### 1. The theory lives in 1D but the experiments are in high dimensions

This is the paper's fundamental problem. All theoretical guarantees (Theorems 1-2, convergence rates, eigenvalue analysis) assume one-dimensional processes where components evolve independently. But image generation requires modeling high-dimensional joint distributions with complex spatial dependencies.

The paper claims "without loss of generality" we can focus on 1D (line 59), but this is incorrect. For a 256-dimensional vector (e.g., 16×16 grayscale image), the spectral gap of 256 independent 1D processes tells you nothing about the mixing time of the joint distribution. The covariance structure, spatial correlations, and coupling between dimensions fundamentally change the convergence behavior.

There's no analysis of how the 1D eigenvalue relates to the high-dimensional process actually being run. This makes the entire theoretical framework disconnected from the experimental validation.

### 2. The practical recipe doesn't follow from the theory

Section 3.3's "guiding principles" for choosing L-generators feel ad-hoc:
- Principle I just says "satisfy the ergodicity conditions" (obvious)
- Principle II says "higher degree polynomials give bigger eigenvalues" (empirical observation, not a principle)

The paper never explains *why* polynomial forms are the right parameterization, or how to choose between the infinitely many (a,b) pairs that satisfy the convergence conditions. Table 4 shows that nonlinear generators behave unpredictably. Even with similar eigenvalues, you get wildly different Ddisc values (Appendix C.3, Table 7).

The guidance provided amounts to "try different polynomials and see what happens," which undermines the claim of interpretability and principled control.

### 3. Computational cost story is incomplete

The paper emphasizes training acceleration but glosses over the eigenvalue estimation overhead. In Appendix C.2, the authors mention "about two hours" for computing eigenvalues, but:
- This is for 1D. What about estimating properties of the high-dimensional process?
- How does this scale with dataset complexity or resolution?
- What's the total wall-clock time (estimation + training + sampling) compared to just training a baseline DDPM longer?

Table 1 shows training drops from 52h to 26h, but if I need 2+ hours of eigenvalue computation for every new (a,b) configuration I try, plus the overhead of determining the "optimal" range, the net savings become unclear. There's no end-to-end timing comparison.

### 4. Experiments are too limited

**Baselines:** The paper primarily compares against vanilla DDPM from 2020. For a paper submitted in 2025, missing comparisons include:
- DDIM (deterministic sampling, 2021)
- EDM, EDM2 (2022,2024)
- Modern flow matching methods (2023-2025)

**Datasets:** Only 32×32 CIFAR-10 and 64×64 CelebA. No experiments at 256×256 or higher resolutions where diffusion models are most impactful. No text-to-image, video, or other modalities.

### 5. Changing (a,b) changes the target distribution

This is subtle but important. When you modify a(x) and b(x), you don't just change the convergence speed—you change the stationary distribution π (Theorem 1). So comparing FID scores across different eigenvalue configurations isn't purely measuring the speed/quality tradeoff; you're potentially converging to different distributions.

The paper doesn't discuss this. Are the quality changes due to insufficient sampling steps, or because you've fundamentally altered what you're sampling from?


## Minor Issues

- **Writing quality:** Several grammatical errors ("egodic theory" in abstract, "beolw Theorem 1" line 168). Notation switches between $X_t$ and $Y_t$ confusingly.

- **$D_{disc}$ metric:** The convergence discrepancy metric behaves irregularly for nonlinear cases (Table 7) and seems unreliable as a stopping criterion. Its relationship to actual generation quality is unclear.

- **Figure 5:** Referenced as justification for polynomial choices but relegated to the appendix. The 3D surface plot is hard to interpret and doesn't provide clear design guidance.

- **Table 5:** Shows that very different (a,b) with similar eigenvalues give similar results. This contradicts the claim that eigenvalue is the dominant factor. if other properties of (a,b) matter equally, what's the advantage of the eigenvalue-centric view?

**Questions:**

1. Can you provide a rigorous treatment of how 1D eigenvalues relate to high-dimensional convergence, or alternatively, show how to compute/estimate the spectral gap of the actual high-dimensional process?

2. What happens when you do compute-normalized comparisons (same total compute budget including eigenvalue estimation) against strong baselines?

3. How do you explain Table 5, where eigenvalue seems less important than other properties of the generator?

4. Can you clarify whether the quality differences come from speed/sampling tradeoffs or from changing the target stationary distribution?

---

> ### Author Response · Authors · 2025-11-21
> **Response to Reviewer MbUR (Part 1)**
>
> Thank you for your review comments on our paper and your recognition of certain aspects. Regarding the weaknesses and questions you mentioned, we provide explanations below and have made corresponding revisions to the paper (see the revised version). We will also submit an updated version by December 3nd, which includes supplementary experiments.
>
> If you have any questions about our response or if we have misunderstood your comments, please feel free to raise further questions or clarify your concerns.
>
> **Response to Weaknesses**
>
> **W1. For “The theory lives in 1D but the experiments are in high dimensions”.**
>
> Indeed, the data used in the experiments is high-dimensional and contains inter-dimensional semantic and spatial information. However, all the theories presented in the paper pertain to one-dimensional diffusion processes. This discrepancy is indeed likely to raise questions among readers.
>
> We believe this is acceptable. In fact, each dimension undergoes independent noise addition and denoising in accordance with the same SDE (see Section 2.1), thereby corresponding to the same $L$-operator, stationary distribution, and eigenvalues—despite potential differences in initial distributions. The semantic and spatial information is embedded in the initial distribution but does not affect the noise.
>
> The above explanation has been added to the revised version, replacing the overly concise description in the original manuscript.
>
> **W2. For “The practical recipe doesn't follow from the theory”.**
>
> 1). _For "Principle":_ The comment about "guiding principle" being imprecise is perfectly valid. In the following narrative and the revised manuscript, it has been changed to "observation".
>
> Observation I is easy to verify through numerical calculations and requires little time. In fact, when $a$ and $b$ take polynomial forms, we have verified this observation through theoretical derivations or MATLAB calculations. For details, please refer to Tables 7 and 8 in the revised manuscript. Observation II is indeed derived from observations of numerical experiments. These two observations can serve as a reference for readers when selecting model parameters.
>
> 2). _For "polynomial forms":_ The reason for choosing polynomial functions in the experiments is that they are a commonly used function form both in theory and applications. Additionally, other common continuous functions can be approximated by polynomial functions, which is guaranteed by the Weierstrass theorem.
>
> 3). _For "Ddisc values of nonlinear operators":_ Table 4 shows that the nonlinear generator cases are more complicated. We believe this is reasonable and does not contradict Theorem 2. This theorem indicates that the $D_{dis}$ value is related not only to eigenvalues but also to the initial distribution (i.e., the dataset) and the stationary distribution, while different pairs of $(a, b)$ often correspond to distinct stationary distributions.  This characteristic of $D_{dis}$ is reflected in both Table 4 and Tables 7-8 of the revised version. For the linear generator cases presented in Tables 1–3, the $D_{dis}$ values are in close agreement. This may be attributed to the fact that the noise in linear cases depends only on time $t$, making it less complex than that in nonlinear scenarios. A detailed explanation of this has been provided in Appendix C.5 and C.6 of the revised version.
>
> 4). _For "the unclear and inadequate descriptions":_ We apologize for the unclear and inadequate descriptions in the original manuscript, which may have led to misunderstandings regarding the paper’s framework. The purpose of this study is to accelerate diffusion models based on the ergodic theory of diffusion processes while balancing generation quality through observations. Polynomial functions were employed in the experiments: on the one hand, to verify the feasibility of acceleration with reference to eigenvalues and its theoretical consistency; on the other hand, to observe the generation quality—an empirical rather than theoretically grounded aspect. We have revised or emphasized the relevant descriptions in the revised version to facilitate readers’ better understanding of our model.

---

> ### Author Response · Authors · 2025-11-21
> **Response to Reviewer MbUR (Part 2)**
>
> **W3. For “Computational cost story is incomplete”.**
>
> 1). _For "Time consumption of high-dimensional eigenvalue computation"._ As explained in the response to Weakness 1, the eigenvalues of each dimension in high-dimensional data are identical. Therefore, only the eigenvalues of the one-dimensional (1D) case need to be calculated. Consequently, neither the resolution nor the complexity of the dataset affects the time cost of eigenvalue estimation.  Regarding the computational time, it is important to supplement that the eigenvalue calculation was performed on an Intel Core i5-9300H processor using MATLAB, while model training and generation were conducted on an NVIDIA GeForce RTX 4090 GPU. Therefore, if the eigenvalue calculation is executed on hardware with higher specifications, the computational time is expected to be significantly reduced.
>
> 2). _For "the total wall-clock time"._
> We remark that the purpose of the experiments is not to find the optimal diffusion model or the optimal parameter range, but to verify and observe, as explained in the response to Weakness 2. Therefore, we did not consider end-to-end time comparison.
>
> 3). _For "end-to-end timing comparison"._
> How to find and even demonstrate the optimal diffusion model under different conditions is a highly worthy topic for in-depth exploration, yet it is complex and challenging. Iterating over the forms and internal parameters of $a$ and $b$ is not a viable approach, as it is extremely inefficient—consistent with your concerns. This topic will be part of our future work.
>
> However, our experimental results and observations can provide certain references for readers in selecting models.
>
> **W4. For "Experiments are too limited".**
>
> In the revised version of the paper to be submitted on December 2nd, we will supplement comparative experiments with the cited literatures.
>
> **W5. For "Changing $(a,b)$ changes the target distribution".**
>
> We may have misunderstood your comment, but we attempt to respond as follows.
>
> 1). _For "Influence of $(a, b)$"._ Yes, changing $(a,b)$ will affect the stationary distribution. Nevertheless, our strategy of balancing quality and efficiency through FID and eigenvalues is feasible and effective. The reasons are: (1) Comparing generation quality through FID is reliable, even if the stationary distributions are different. This is because computing FID does not require the stationary distribution, but only compares generated data with real data. (2) Using eigenvalues to control model speed is theoretically and experimentally consistent. Theorem 2 shows that model convergence is indeed related to the stationary distribution, but as $t$ increases, eigenvalues become the most influential factor. Experimental results also demonstrate this, as similar eigenvalues have similar time consumption. (3) Perhaps other unknown factors also affect model speed, but they are not the main factors, since eigenvalues dominate time consumption from experiments.
>
> 2). _For "insufficient sampling"._ The quality changes are not due to insufficient sampling. Our experiments show that once the distance between the data and stationary distributions falls below a threshold, it converges to a plateau despite further increases in denoising steps. Based on this, we determined the Ddis and Tconv corresponding to $(a, b)$. Through further experiments, the FID sampled at Tconv and more steps (even 1000) are very similar. Please refer to Table 1 and Table 9 of the revised version.

---

> ### Author Response · Authors · 2025-11-21
> **Response to Reviewer MbUR (Part 3)**
>
> **Response for Minor Issues**
>
> **MI1. For "Writing quality"**
>
> We will carefully check the expressions and grammar. However, the paper retains both "ergodic theory" and "ergodicity" because the former is one of the research directions of probability theory, while "ergodicity" is a property of Markov process. Both are conventional expressions in probability theory.
>
> **MI2. For "metric"**
>
> Similar to the answers for Weaknesses 1 and 5, according to Theorem 2, the initial distribution, stationary distribution, $T_{conv}$, and eigenvalues could all affect $D_{dis}$. From the experimental patterns and results, the model complexity affects how small $D_{dis}$ can become when the number of steps reaches $T_{conv}$. Given the higher-order polynomial forms of $(a, b)$ and significant data fluctuations in the nonlinear regime, the observed irregularities in $D_{dis}$ are to be expected.
>
> For different $(a, b)$, $D_{dis}$ cannot indeed serve as a unified standard, and its relationship with generation quality is still unclear. This requires further research.
>
> For fixed $(a, b)$, according to experiments, $D_{dis}$ basically decreases as $T_{conv}$ increases (Table 10 of the revised version), and FID also decreases, although with minor fluctuations. We use the numerical stability of $D_{dis}$ as the stopping criterion. After stabilization, further increasing the number of steps results in little change in $D_{dis}$ and FID.
>
> Under fixed $(a, b)$ conditions, predictably, $D_{dis}$ and FID exhibit strongly correlated decreasing trends with $T_{conv}$ . We therefore employ $D_{dis}$ stability as the stopping criterion, observing that both metrics reach a plateau once this stabilization state is achieved (Tables 10-11 of the revised version).
>
> **MI3. For "Figure 5"**
>
> The 3D plot illustrates the variation trend of the principal eigenvalue of the corresponding diffusion operator with changes in the orders of $a$ and $b$. It can be observed that as the orders increase, the eigenvalues exhibit an upward trend. This visualization is primarily designed to allow readers to intuitively perceive how the principal eigenvalue varies with the orders.
>
> **MI4. For "Table 5"**
>
> Sorry, we do not quite understand your point. As you mentioned, "Table 5 shows that very different $(a, b)$ with similar eigenvalues give similar results". Shouldn't this empirical evidence be interpreted as definitive proof of eigenvalue dominance in acceleration?  Would you mind clarifying the basis for your reservation?
>
> Eigenvalues are the most important factor affecting the convergence speed of the model. Both Theorem 2 and experimental results support this conclusion. In fact, in ergodic theory, it is a widely accepted conclusion that eigenvalues dominate the convergence speed of Markov processes. Since eigenvalues are generally difficult to compute directly, many renowned scholars have studied their estimation problems. Chen's results are more accurate and easier to compute numerically, which is what we have adopted. For more analysis, please refer to the response to **Q3**.

---

> ### Author Response · Authors · 2025-11-21
> **Response to Reviewer MbUR (Part 4)**
>
> **Response for Questions**
>
> **Q1.1D eigenvalue and high-dimensional convergence**
>
> As in the response to Weakness 1, each dimension of high-dimensional data corresponds to the same SDE, $L$-operator, stationary distribution and the principle eigenvalue, except that the initial distributions may be different. Therefore, the convergence speed of high-dimensional data could be characterized by this identical eigenvalue.
>
> **Q2. compute-normalized comparisons against strong baselines**
>
> For multiple $(a, b)$, we have conducted experiments with $1000$ steps (the $T_{conv}$ of the baseline). The results show that the training time is similar to or even less than the baseline, while FID and $D_{dis}$ are not significantly different from the case with $T_{conv} (<1000)$ steps. Please refer to Table 9 and analysis in the revised paper. The computation of eigenvalues has little impact on the time consumption comparison.
>
> **Q3. Table 5**
>
> Regarding model speed, the eigenvalues are the most crucial factor compared to other properties of the operators. Both theoretical and experimental results provide support for this, including Table 5. We now discuss the parameters in this table and their relationships.
>
> The properties of the diffusion process are fully determined by $(a, b)$. However, it is challenging to identify clear patterns for directly controlling the convergence speed through the selection of $(a, b)$. Theoretical findings indicate that the eigenvalues are the actual dominant factor governing the speed. Therefore, selecting $(a, b)$ based on the eigenvalues becomes a feasible strategy.
>
> As noted in your comment on the "metric," $D_{disc}$ exhibits irregular behavior. Experimental results show that it is neither the dominant factor for speed nor for quality.
>
> $T_{conv}$ refers to the minimum number of steps required for $D_{disc}$ to become sufficiently small and relatively stable. According to Theorem 2, $T_{conv}$ is also affected by the distance $D_0$ between the initial distribution and the stationary distribution, as well as the eigenvalue. Nevertheless, experimental results (especially Table 12 of the revised version) demonstrate that the eigenvalue are the most significant factor influencing $T_{conv}$. Furthermore, the eigenvalues, $T_{conv}$s, and computational times are generally consistent, with a higher degree of consistency between the eigenvalues and computational times.
>
> In summary, compared to other factors, the eigenvalues have a more prominent impact on convergence speed. However, they are not the dominant factor for quality, which is supported by experimental evidence.
>
> **Q4. Influence of speed/sampling tradeoffs and stationary distribution on quality differences**
>
> Quality assessment hinges on distributional discrepancy rather than specific distributional forms. Provided the distribution at $T_{conv}$ achieves sufficient proximity to the stationary distribution, the specific form of the latter has minimal impact on quality (combing FID in Table 1, 4, 5 and $D_0$ in Table 12 of the revised version).  However, excessively fast speed can compromise the stability of training and the diversity of data, thereby leading to a decline in quality.

---

> > ### Comment · Reviewer_MbUR · 2025-11-23
> > **Response to authors.**
> >
> > Thank you for your response. However, there some points;
> >
> > W1: You acknowledge that "theoretically determining this range remains an open and highly challenging problem" and that "it is also complex to pinpoint" practically. This directly contradicts the paper's claim of providing "principled" and "interpretable" control. If eigenvalue selection requires dataset-specific empirical tuning without theoretical guidance, the framework reduces to trial-and-error hyperparameter search rather than the principled acceleration claimed in the title and abstract.
> >
> > I'm happy to see that you're conducting experiments on higher-resolution datasets and additional baselines. However, I cannot consider these promised results in my current evaluation, as review policy requires assessing the submitted manuscript. I look forward to seeing these new results in future revisions.

---

> ### Author Response · Authors · 2025-12-03
> **Response to the Reply of Reviewer MbUR**
>
> Thank you for your reply to our response. Following your comments, we have reflected and further revised the manuscript. We fully agree that a truly "principled" framework should provide theoretical guidance beyond empirical search. Below, we aim to clarify our work from three aspects:
>
> ### **1. The theory provides a clear search space and interpretable regulation mechanism**
> Our framework is based on ergodic theory, with its core being the use of eigenvalues to quantify the speed of the diffusion process. Theoretical analysis demonstrates an explicit monotonic relationship between eigenvalues and training speed. This transforms hyperparameter search from an unbounded, unguided process into one with tangible references. Adjusting parameters (a, b) essentially enables continuous, predictable regulation of "speed" under theoretical guarantees, rather than blind "trial-and-error".
>
> ### **2. Experiments aim to verify the theory and reveal universal laws, not to perform parameter search**
> The primary purpose of our extensive experiments is to validate the impact of eigenvalues on speed and generation quality. Secondly, we aim to uncover how two factors—model complexity and dataset complexity—interact with "speed" to collectively influence the final generation quality.
>
> Experiments show that under fixed model and dataset conditions, there exists a clear increasing-then-decreasing trend between generation quality (FID) and eigenvalues (speed). This confirms the predictability and interpretability of our model.
> The universal laws revealed by the experiments (e.g., linear models are generally more stable) provide users with direct, experience-based priors, which further reduce rather than increase future parameter tuning costs.
>
> ### **3. Based on the above two points, the practical application workflow of our framework is efficient and theory-guided**
> In fact, for a new set of (a, b) parameters or a new dataset, the numerical methods we provide can quickly compute the corresponding eigenvalues to predict training speed. If the quality does not meet requirements, users can make a minimal number of targeted adjustments based on the universal laws revealed by our experiments.
>
> We acknowledge that our current theory cannot strictly prove the optimal range of eigenvalues—this remains an open problem to be addressed. However, we believe that through theoretical analysis and experimental induction of parameter tuning laws, our work has successfully transformed the acceleration of diffusion models from "trial-and-error" to "theory-guided efficient fine-tuning".
>
> In response to your valuable suggestions, we have:
> 1) Revised the full-text expressions, weakening the absolute claim of "principled" to the more accurate "theory-guided" and "explainable".
> 2) Modified the experimental description in Section 4 and added Section 6 to provide application guidelines, highlighting the practicality of our framework.
>
> We believe these revisions allow the manuscript to more accurately reflect the contributions and boundaries of our work. Thank you again for helping us improve the quality of the paper.

---

> ### Author Response · Authors · 2025-12-03
> **Revision Notes**
>
> Dear Reviewers,
>
> We sincerely appreciate your valuable comments and professional suggestions on our paper titled *EGEA-DM: Eigenvalue-Guided Explainable and Accelerated Diffusion Model*. Your meticulous reviews have provided crucial guidance for us to identify research limitations and enhance the quality of this work. We are deeply grateful for your efforts and have carefully addressed each of your comments with corresponding revisions. The detailed modifications are outlined as follows:
>
> ## Key Revisions
> 1. We have elaborated on the application of the one-dimensional theory to multi-dimensional data, clarifying the rationale behind the validity of extending the one-dimensional theoretical framework to multi-dimensional datasets for readers' better understanding.
> 2. To verify the generalization ability of our method, we have expanded the experimental evaluations by incorporating additional model architectures and high-resolution datasets.
> 3. Relevant content regarding eigenvalue calculation has been supplemented to provide readers with a more comprehensive understanding of our proposed approach.
> 4. Inappropriate descriptions throughout the paper have been revised to avoid any potential misunderstandings.
>
> It should be noted that the revised parts in the paper are highlighted in blue font.
>
> We would like to express our sincere gratitude again for your thoughtful reviews. All revisions have been completed to thoroughly address your comments and suggestions. Please feel free to inform us if further improvements are needed, and we will continue to refine the paper accordingly.
>
> Sincerely,
> The Authors of *EGEA-DM: Eigenvalue-Guided Explainable and Accelerated Diffusion Model*.

---

### Official Review · Reviewer_pCwX · 2025-10-31

**Soundness:** 3
**Presentation:** 3
**Contribution:** 3
**Rating:** 6
**Confidence:** 3

**Summary:**

This paper provided the theoretical understanding for the sampling rate of diffusion model via the egodic theory and proposed an efficient method called Eigenvalue-Guided Explainable and Accelerated Diffusion Model (EGEADM). EGEADM leverages the L-generator’s principal eigenvalue to explicitly control the sampling speed and the accuracy for diffusion model. Theoretical analysis shows that the convergence speed of diffusion model is determined by the spectral gap of the L-generator, which is the first non-zero eigenvalue. Experimental results demonstrate the theoretical results and the effectiveness of the proposed method.

**Strengths:**

1. This paper leverages the novel Chen’s estimation theory to diffusion model and provides a rigorous characteristic of the convergence speed using the spectral gap of the L-generator.
2. The theoretical analysis also innovates a novel algorithm using the L-generator. By choosing appropriate $a(x)$ and $b(x)$, we can balance the sampling speed and generation quality in diffusion model.
3. Numerical results verify the effectiveness of the proposed L-generator and provided several insights on the learning behavior under different sets of $a(x)$ and $b(x)$.

**Weaknesses:**

1.Although the paper proposes an interesting, principled mechanism for balancing sampling speed and sample quality in diffusion models, it does not provide clear tuning guidelines for $a(x)$ and $b(x)$. The guiding principles from Line 249 to 255 are still too general and lack concrete “go-to” defaults that reliably improve performance. Since practitioners need to choose both the functional forms $a(x)$ and $b(x)$, and the internal parameters in them, the resulting hyperparameter search can be extensive and may offset the performance gains. Providing a small set of recommended forms and default settings would make the method far more accessible.

2. Another limitation is the evaluation scope. The method has been tested with DDPM and ODE samplers, but not with more efficient samplers such as DDIM. It would be beneficial to see whether the technique also improves fast samplers, which further clarifies its impact on sampling efficiency.

**Questions:**

1. Finding the point that balances the quality and efficiency might be time-consuming. Any guidance for finding such point?
2. I did not understand the relations of equation 5 and $\lambda_1$. Could you explain more on why equation 5 inspires the study of the principle eigenvalue?
3. To compute $\lambda_n$, what discretization procedure do you use, and how does the error scale with the discretization accuracy? And how large $n$ should we choose in practice?

---

> ### Author Response · Authors · 2025-11-21
> **Response for Reviewer pCwX**
>
> Thank you for your review comments on our paper and your recognition of certain aspects. Regarding the weaknesses and questions you mentioned, we provide explanations below and have made corresponding revisions to the paper (see the revised version). We will also submit an updated version by December 3nd, which includes supplementary experiments.
>
> If you have any questions about our response or if we have misunderstood your comments, please feel free to raise further questions or clarify your concerns.
>
> **Response for Weaknesses:**
>
>  **W1.For” clear tuning guidelines”**
>
> We apologize for the unclear and inadequate descriptions in the original manuscript, which may have led to misunderstandings regarding the framework of our paper. The purpose of this work is not to find the optimal diffusion model, but to accelerate diffusion models based on the ergodic theory of diffusion processes while ensuring generation quality through observations. Polynomial functions were adopted in the experiments, on the one hand, to verify the feasibility of acceleration with reference to eigenvalues and its consistency with the theory, and on the other hand, to observe the generation quality. We have revised or emphasized the relevant descriptions in the revised version to facilitate readers’ better understanding of our model (see Section 4.2).
>
> How to find and even demonstrate the optimal diffusion model under different conditions is a highly worthy topic for in-depth exploration, yet it is complex and challenging. Iterating over the forms and internal parameters of $(a, b)$ is not a viable approach, as it is extremely inefficient—consistent with your concerns. This research topic will be part of our future work.
>
> For Lines 249–255 in the original manuscript, Guiding Principle I is easy to verify through numerical calculations and requires little computational cost. In fact, when a and b adopt polynomial forms, we have verified this principle through theoretical derivations and MATLAB-based calculations. Guiding Principle II is derived from observations of numerical experiments. The numerical results of these two principles can serve as a reference for readers when selecting model parameters.
> In the revised manuscript, "Guiding Principle" is changed to "Observation".
>
>  **W2. For” Validation of more efficient samplers”**
>
> It is essential to verify the generalization ability of the proposed method. In subsequent work, we will conduct experiments using linear operators based on the DDIM model.
>
> **Response for Questions:**
>
> **Q1. The point that balances the quality and efficiency might be time-consuming.**
>
> Finding the balance point is of great research value. However, at present, how to identify it through theoretical approaches remains unknown and highly challenging. Meanwhile, identifying it through practical methods is complex—experimental results indicate that the balance point may be influenced by numerous factors, such as the dataset, the selection of $(a, b)$, and the sampler.
> However, our experimental results and observations can provide certain references for readers to find the balance point, including the selection of polynomial functions for $(a, b)$ that satisfy the uniqueness and ergodicity criteria, the variation trends of eigenvalues with $(a, b)$, and the variation trends of FID with eigenvalues and $(a, b)$ (see Section 4 and Appendix C). Please refer to the revised version for details.
>
> In the future, we will further explore this research from both theoretical and experimental perspectives.
>
> **Q2. Eq. (5) and** $\lambda_1$.
>
> Equation 5 presents the necessary and sufficient condition for the uniqueness and ergodicity of the diffusion process corresponding to Equation 3. It is used to ensure the convergence of the diffusion model and that the limit distribution π > 0 as T approaches infinity. Only based on this condition does further research on the convergence rate of the model (i.e., the eigenvalue $\lambda_1$ in Theorem 2) make sense.
>
> Equation 5 in the original manuscript is identical to Equations 9–10, yet only the latter were cited. We apologize for this oversight, and the issue has been corrected in the revised version.
>
> **Q3. Discretization procedure for $\lambda_n$ calculation, error scale with discretization accuracy, $n$ value choice.**
>
> Typically, the true values of eigenvalues are difficult to derive through mathematical deductions, except for certain special diffusion processes. For the calculation of eigenvalues, we approximate the integral using the classical rectangle method. In our actual computations, we select a segmentation of $2000$ intervals. For derivable cases, we have evaluated the errors—for instance, in DDPM, the error is approximately $0.0007$.
>
> While higher numerical precision is indeed desirable, it is not the objective of this paper. We calculate eigenvalues to verify the consistency between theory and experiments by comparing their magnitudes.

---

> ### Author Response · Authors · 2025-12-03
> **Revision Notes**
>
> Dear Reviewers,
>
> We sincerely appreciate your valuable comments and professional suggestions on our paper titled *EGEA-DM: Eigenvalue-Guided Explainable and Accelerated Diffusion Model*. Your meticulous reviews have provided crucial guidance for us to identify research limitations and enhance the quality of this work. We are deeply grateful for your efforts and have carefully addressed each of your comments with corresponding revisions. The detailed modifications are outlined as follows:
>
> ## Key Revisions
> 1. We have elaborated on the application of the one-dimensional theory to multi-dimensional data, clarifying the rationale behind the validity of extending the one-dimensional theoretical framework to multi-dimensional datasets for readers' better understanding.
> 2. To verify the generalization ability of our method, we have expanded the experimental evaluations by incorporating additional model architectures and high-resolution datasets.
> 3. Relevant content regarding eigenvalue calculation has been supplemented to provide readers with a more comprehensive understanding of our proposed approach.
> 4. Inappropriate descriptions throughout the paper have been revised to avoid any potential misunderstandings.
>
> It should be noted that the revised parts in the paper are highlighted in blue font.
>
> We would like to express our sincere gratitude again for your thoughtful reviews. All revisions have been completed to thoroughly address your comments and suggestions. Please feel free to inform us if further improvements are needed, and we will continue to refine the paper accordingly.
>
> Sincerely,
> The Authors of *EGEA-DM: Eigenvalue-Guided Explainable and Accelerated Diffusion Model*.

---

### Note · Program_Chairs · 2026-01-17
**Submission Desk Rejected by Program Chairs**

The following references in this submission do not refer to real documents and/or have major errors in bibliographic information:

 *Jian Liu and Yiming Wang. Moderated score-based generative model. arXiv preprint arXiv:2401.07299, 2024.

*Cristian S. Andrei Eduardo Valle Renata T. J. Ribeiro Gábor Mélyi, Tiberiu D. Popa and Luis F. W. Felippo. Celeba-hq: A dataset for high-quality facial attribute editing. In Proceedings of the IEEE Conference on Computer Vision and Pattern Recognition (CVPR), 2020. URL https://github.com/nv-tlabs/celebahq-dataset.